# Psychological distress and health-related quality of life in patients after hospitalization during the COVID-19 pandemic: A single-center, observational study

Johan Hendrik Vlake [1,2]*, Sanne Wesselius [1], Michel Egide van Genderen[1,2], Jasper van Bommel[2], Bianca Boxma-de Klerk[3], Evert-Jan Wils[1]

1 Department of Intensive Care, Franciscus Gasthuis & Vlietland, Rotterdam, The Netherlands,
2 Department of Intensive Care, Erasmus Medical Centre, Rotterdam, The Netherlands, 3 Department of Statistics and Education, Franciscus Academy, Franciscus Gasthuis & Vlietland, Rotterdam, The Netherlands

* h.vlake@franciscus.nl

## Abstract

### Introduction

Illnesses requiring hospitalization are known to negatively impact psychological well-being and health-related quality of life (HRQoL) after discharge. The impact of hospitalization during the Severe Acute Respiratory Syndrome Corona Virus 2 (SARS-CoV-2) pandemic on psychological well-being and health-related quality of life is expected to be higher due to the exceptional circumstances within and outside the hospital during the pandemic surge. The objective of this study was to quantify psychological distress up to three months after discharge in patients hospitalized during the first coronavirus disease 2019 (COVID-19) pandemic wave. We also aimed to determine HRQoL, to explore predictors for psychological distress and HRQoL, and to examine whether psychological distress was higher in COVID-19 confirmed patients, and in those treated in Intensive Care Units (ICUs).

### Methods

In this single-center, observational cohort study, adult patients hospitalized with symptoms suggestive of COVID-19 between March 16 and April 28, 2020, were enrolled. Patients were stratified in analyses based on SARS-CoV-2 PCR results and the necessity for ICU treatment. The primary outcome was psychological distress, expressed as symptoms of post-traumatic stress disorder (PTSD), anxiety, and depression, up to three months post-discharge. Health-related quality of life (HRQoL) was the secondary outcome. Exploratory outcomes comprised predictors for psychological distress and HRQoL.

### Results

294 of 622 eligible patients participated in this study (median age 64 years, 36% female). 16% and 13% of these patients reported probable PTSD, 29% and 20% probable anxiety, and 32% and 24% probabledepression at one and three months after hospital discharge,

**Data Availability Statement:** All relevant data are within the manuscript and its Supporting Information files.

**Funding:** The author(s) received no specific funding for this work.

**Competing interests:** The authors have declared that no competing interests exist.

respectively. ICU patients reported less frequently probable depression, but no differences were found in PTSD, anxiety, or overall HRQoL. COVID-19 patients had a worse physical quality of life one month after discharge, and ICU patients reported a better mental quality of life three months after discharge. PTSD severity was predicted by time after discharge and being Caucasian. Severity of anxiety was predicted by time after discharge and being Caucasian. Depression severity was predicted by time after discharge and educational level.

## Conclusion

COVID-19 suspected patients hospitalized during the pandemic frequently suffer from psychological distress and poor health-related quality of life after hospital discharge. Non-COVID-19 and non-ICU patients appear to be at least as affected as COVID-19 and ICU patients, underscoring that (post-)hospital pandemic care should not predominantly focus on COVID-19 infected patients.

## Introduction

During the first coronavirus disease 2019 (COVID-19) peak in the Netherlands between March and May 2020, extensive measures were taken to reduce the spread of the virus and to safeguard medical care; in general, it was advised to work from home, minimize social contacts, keep distance in public places, and to remain in self-quarantine when experiencing symptoms suggestive of COVID-19 [1]. Upon hospital admission, patients suspected of COVID-19 were strictly isolated, visitation was restricted, and contact with healthcare workers was limited. Moreover, COVID-19 and its consequences were a frequent subject in the media, and patients were as such constantly reminded of the possible severity of the disease.

Illnesses requiring hospitalization, particularly those requiring intensive care unit (ICU) treatment, are known to negatively impact post-discharge psychological well-being; symptoms of depression and anxiety occur in up to 67% of hospitalized patients and symptoms of post-traumatic stress disorder (PTSD) in up to 45% [2–15]. This post-hospitalization psychological distress negatively impacts the health-related quality of life (HRQoL) [7,11,12,14]. Both demographic characteristics, such as female gender, lower educational level, unemployment and non-western ethnicity, and treatment-related characteristics, such as duration of admission and severity of the disease, are known to be negatively associated with psychological well-being of patients [2,8,16–19].

The uncertainties and measures surrounding the pandemic raised concerns of increased psychological distress in non-hospitalized citizens [20]. Since measures taken in hospital were more drastic, concerns of the mental well-being of hospitalized patients were even higher [21,22]. Data from the previous SARS and MERS epidemics support this concern and early studies suggest that up to 50% of COVID-19 patients suffer from psychological distress up to two months after hospital discharge [23–28]. More recently, Taquet et al. estimated that the risk for psychiatric sequelae is higher in COVID-19 patients and in those admitted to ICU using electronic health records data. Complementary observational studies are needed to add direction as to whether these risks may be attributed to COVID-19 or hospitalization during a pandemic, and corroborate on the different risks between specific subsets of patients, such as non-COVID-19 and ICU patients [29].

In this study, we first aimed to quantify psychological distress up to three months post-discharge in patients hospitalized during the first pandemic peak with symptoms suggestive of

COVID-19. Additionally, we assessed their HRQoL, explored predictors for psychological distress and HRQoL, and examined whether psychological distress was more prevalent or more severe in COVID-19 confirmed patients, or in those treated in ICU.

## Materials and methods

### Ethics approval

The study protocol was approved by the Institutional Review board of the Franciscus Gasthuis & Vlietland and deemed not to fall under the Medical Research Involving Human Subjects Act (WMO) (S1. Registered study protocol). The need for written informed consent was waived. Patients in our hospital are actively informed about the use of their anonymous data in research activities and can object against the use their data. No data were used of patients who objected against this use. The study was registered at TrialRegister.nl (registration number: NL8882; S1 File. Study protocol). The reporting of this study follows the Strengthening the Reporting of Observational Studies in Epidemiology (STROBE) guideline (S2 File. STROBE Checklist) [30].

### Study design and setting

This single-center, observational cohort study was conducted in the Franciscus Gasthuis & Vlietland hospital in Rotterdam, the Netherlands, from March 16 to September 14, 2020. This period coincided with the first COVID-19 peak in the Netherlands. During this period, several protective measures were taken in the hospital, such as the prohibition of visiting hospitalized patients and strict isolation of suspected patients until COVID-19 was ruled out.

### Participants

Eligible patients were aged ≥18 years and hospitalized between March 16 and April 28, 2020, with symptoms suggestive of COVID-19, defined as the presence of respiratory symptoms (dyspnea, coughing, sore throat, rhinorrhea, saturation <94% or respiratory rate >24/minute) and/or gastro-intestinal symptoms (diarrhea or vomiting) with a duration ≥24 hours, and who survived until one month after hospital discharge. Patients who were unable to understand the Dutch language or did neither have a formal home address nor e-mail address were excluded.

### Procedures

Patients were approached one month post-discharge by sending an information letter and the first questionnaires. Patients were asked to either send back the questionnaires filled out to participate, or to return the questionnaire blank to decline participation. Patients who did not return the questionnaires were contacted twice by telephone as a reminder. A second set of questionnaires was sent three months post-discharge to participating patients, to patients who did not return the first set of questionnaires and could not be reached by telephone, and to patients who had consented to participate in the second questionnaire by phone. The last patient was discharged June 14, 2020, and follow-up lasted until three months after (September 14, 2020). Convenient sampling was used. All patients admitted between March 16 and April 28, 2020, and who responded to one of the two follow-up assessments were included.

   We randomly approached non-responders (who did not respond at both time-points) four months after hospital discharge, i.e., one month after sending the last questionnaire, and asked them to fill out a single set of questionnaires to analyze non-responder's bias.

## Measures

All data were gathered using the International Severe Acute Respiratory and emergency Infection Consortium (ISARIC, Oxford, United Kingdom) and Franciscus Corona Registry in Castor Electronic Data Capture (Castor EDC, Amsterdam, the Netherlands). The ISARIC database in an international initiative to collect baseline demographics and treatment-related characteristics of all patients admitted to the hospital with respiratory symptoms during the SARS-CoV-2 pandemic. The Franciscus Corona Registry is a local addition to the ISARIC database, in which variables that were not collected in the ISARIC database, but were required for the current trial, were collected. All data was collected by members of the study team (JV and SW).

**Baseline demographics and treatment-related characteristics.** The following baseline demographics were retrieved from electronic healthcare records: age (years), ethnicity (Caucasian, black, Surinamese/Hindustan, Arab (not specified), Turkish, Moroccan, others, or unknown), sex at birth (male or female), body mass index (kg/m2), comorbidities (yes/no: hypertension, chronic cardiac disease, chronic pulmonary disease, asthma, tuberculosis, chronic kidney disease, mild liver disease, moderate liver disease, chronic neurological disease, dementia, chronic hematologic disease, diabetes type I or II, rheumatologic disorder, malignant neoplasm; total number of comorbidities), smoking status (yes, never smoked, former smoker, unknown). Patients were asked about their educational level (i.e., elementary school, high school, intermediate vocational education, higher professional education, university education) and employment characteristics (employed before hospitalization yes/no, weekly work hours before admission, weekly work hours after discharge, healthcare worker yes/no). Additionally, we asked patients about their mental history, i.e., whether they had encountered psychological impairments in the past 5 years and whether they were treated for these impairments by a psychologist, psychiatrist or had received medication. Patients were free to decide whether or not to answer the questions regarding their mental history.

The following treatment-related characteristics were retrieved from electronic healthcare records: cause of admission, treatment restrictions at the day of hospital admission, last registered treatment restriction before discharge, hospital length of stay, ICU admission (yes/no, length of stay), SOFA score at admission, P/F ratio at admission, S/F ratio at admission, oxygen therapy (yes/no, duration), non-invasive ventilation (yes/no, duration), invasive ventilation (yes/no, duration), prone positioning (yes/no, duration), tracheostomy (yes/no) and survival during follow-up.

**Primary outcome: Psychological distress.** The primary outcome was the prevalence and severity of probable PTSD, depression, and anxiety, assessed using validated Dutch translations of the Impact of Event Scale–Revised (IES-R) and the Hospital Anxiety and Depression Scale (HADS) at one and three months post-discharge.

The IES-R is a 22-item questionnaire that quantifies the subjective distress a person is experience after a traumatic event [31]. The IES-R yields a sum score, ranging from 0–88 (higher scores indicating more severe symptoms), and subscale scores can be calculated for symptoms of intrusion, avoidance, and hyperarousal. The IES-R sum score was considered as the severity of PTSD, and an IES-R sum score above 24 was defined as probable PTSD [32].

The HADS is commonly used to determine the levels of anxiety and depression that a person is experiencing [33]. The HADS is a 14-item scale that generates ordinal data. Seven of the items relate to anxiety and seven relate to depression. The HADS yields a depression and anxiety sum score, ranging from 0 to 21, with higher scores indicating more severe symptoms. The HADS anxiety and depression score will be considered as the severity of anxiety- and

depression-related symptoms, respectively. A HADS depression or anxiety score above 8 was defined as probable depression or probable anxiety, respectively [33,34].

**Secondary outcome: Health-related quality of life.** The secondary outcome was HRQoL. HRQoL was assessed using validated Dutch translations of the EuroQoL 5-dimensions-5-levels (EQ-5D) and the RAND-36 questionnaires at one and three months post-discharge.

The EQ-5D measures the HRQoL in five dimensions (mobility, self-care, usual activities, pain/discomfort, and anxiety/depression), by which the weight of a health state can be computed into an EQ-5D Time Trade Off (TTO) score. This score ranges from– 0.446 (worst quality of life) to 1.000 (best quality of life) and will be considered as the overall HRQoL. Additionally, patients score their current subjective perceived health state on a visual analogue scale (EQ-5D VAS), ranging from 0 (worst health imaginable) to 100 (best health imaginable) [35,36]. Based on the distribution of age in our cohort, the mean TTO score of the Dutch general population is 0.852 [36]. A TTO score below 0.852 is considered poor, and a TTO score above 0.852 good.

The RAND-36 is a 36-item, self-reported survey of HRQoL, and consists nine scales scores, which are the weighted sums of the questions in their section [37]. Each scale is directly transformed to a scale ranging from 0 (worse score) to 100 (best score) on the assumption that each question carries an equal weight. The nine scores are: physical functioning, social functioning, physical role limitations, emotional role limitations, mental health, vitality, pain, general health, and health change. Based on these scores, a mental and physical component score (MSC-36, PCS-36) can be computed for which the mean in the general population will be 50 with a standard deviation (SD) of 10. The MCS-36 score will be considered as the mental HRQoL and the PCS-36 will be considered as the physical HRQoL. A MCS-36 or PCS-36 score below 50 is considered low, and a MCS or PCS score above 50 is considered good [38].

**Exploratory outcomes.** We additionally explored predictors for the severity and prevalence of psychological distress and the HRQoL. These predictors were chosen based on previous literature, and included: age, gender, ethnicity, educational level, duration of admission, ICU admission, COVID-19 diagnosis and work before admission, and severity of disease in terms of SOFA score at the first day of COVID-19 suspicion (day of enrolment) [2,8,16–19]. Literature supporting these predictors is depicted in S1 Table in the Supporting Information.

## Statistical analysis

Continuous variables are presented as mean, including its standard deviation (SD), if normally distributed, and as median, including its 95% range, if not normally distributed. Categorical variables are presented as absolute and relative frequency. Continuous outcomes include the IES-R sum score, the HADS depression score, the HADS anxiety score, the EQ-5D TTO score, the EQ-5D VAS score, the RAND-36 subscales, the MCS-36, and the PCS-36. The IES-R sum score, HADS anxiety score, and HADS depression score were considered as the severity of PTSD-, depression-, and anxiety-related symptoms, respectively. Categorical outcomes include the prevalence of probable PTSD, probable anxiety, and probable depression. Prevalence of probable PTSD was defined as the proportion patients with an IES-R sum score $\geq$24 and the prevalence of probable anxiety and depression as the proportion of patients with a HADS anxiety or depression score $\geq$8, respectively [32–34]. Psychological distress and HRQoL were assessed one month and three months after hospital discharge to 1) compare results with other research concerning psychological outcomes after hospitalization due to COVID-19 or similar coronaviruses; studies up till now mainly reported data of the first month after discharge [25] and 2) to evaluate the course of psychological symptoms and HRQoL in time after discharge. The internal reliability of all questionnaires used (IES-R,

HADS, EQ-5D and RAND-36) were analyzed using Cronbach's alpha and all showed a high internal reliability (S2 Table).

Patients were stratified based on SARS-CoV-2 PCR outcome, i.e., COVID-19 vs. non-COVID-19, and COVID-19 patients on the necessity for ICU admission, i.e., COVID-19 ICU patients vs. COVID-19 non-ICU patients. Differences between stratifications at baseline were analyzed using a Mann-Whitney U test for continuous variables and using a Fisher's exact test for categorical variables. Differences in outcome measures between stratifications were analyzed using simple linear or logistic regression models, to adjust for at baseline differing characteristics, for continuous and categorical outcomes, respectively. We therefore first performed a simple univariate linear (for continuous outcomes) or logistic (for categorical outcomes) regression analyses, in which all at baseline differing characteristics were analyzed one-by-one. Variables that were associated with the outcome, i.e., a $p$-value $\leq 0.10$, were added as independent variables to the simple multivariate regression models and were as such adjusted for.

To identify possible predictors for the severity and prevalence of psychological distress and the overall, mental, and physical HRQoL, we first conducted univariate mixed effects regression analysis: a linear model for continuous outcomes and a logistic model for categorical outcomes. We used mixed effects regression models to adjust for intergroup (i.e., time) and intragroup differences (i.e., cohort and variables of interest). In these, the possible predictive variable served as independent variable one-by-one and the outcome of interest at both follow-up time points served as dependent variables. Secondly, all variables which showed a $p$-value $\leq 0.10$ in the univariate mixed effects regression model were added to the multivariate mixed effects regression model to determine which variables significantly predicted the outcome. We report the coefficient [95% CI], which implies the estimated mean difference, for linear models, and odds ratios (ORs), including its 95% CI, for logistic models.

Baseline characteristics of all non-responders were compared with those of responders using a Mann-Whitney U test for continuous variables and using a Fisher's exact test for categorical variables. Psychological outcomes of randomly selected non-responders (who did not respond at both time-points) were compared with those of responders, and psychological outcomes of full responders were compared with those of partial responders (i.e., patients who only responded at one or three months), using simple linear or logistic regression models for continuous and categorical variables, respectively. In these, we adjusted for variables that were expected to confound the outcome and differed at baseline between stratifications as described above.

Missing values were not replaced. All analyses were performed using SPSS (Version 27.0) and R for Statistics (R Foundation for Statistical Computing, Vienna, Austria, 2015). A p-value <0.05 was considered statistically significant.

## Results

From March 16 to April 28, 2020, 622 out of 796 patients were admitted with symptoms suggestive of COVID-19 and at least survived up to one month after hospital discharge, of whom 294 patients participated (47%); 252 at one month and 212 at three months post-discharge (**Fig 1**). The response rates per follow-up time point and per questionnaire are depicted in S3 Table. The last patient was discharged June 14, 2020, and follow-up lasted until three months after (September 14, 2020). Non-participating patients either declined participation (n = 261) or did not respond to the questionnaires or to the reminding phone calls (non-responders; n = 67). Responders were less frequently of female gender, had fewer comorbidities, were more frequently smoker, and were more frequently SARS-CoV-2 positive than non-

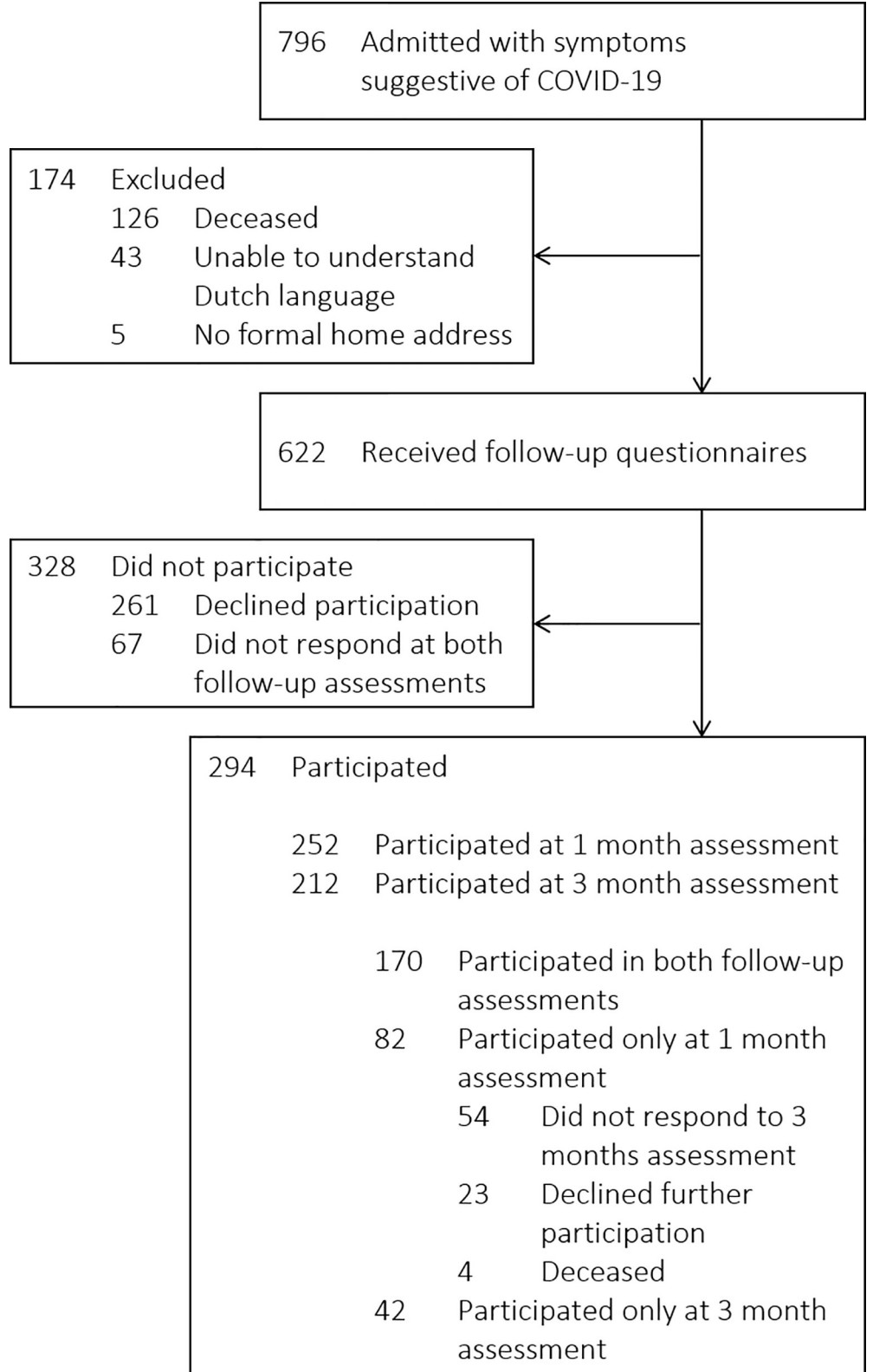

**Fig 1. Flow chart of the recruitment, inclusion, and follow-up of study participants.**

responders (S4 Table). Full responders were more frequently Caucasian, and less frequently Turkish or Moroccan, had more frequently asthma, more frequently had completed university education, were more frequently diagnosed with COVID-19, and were more likely to decease than those who only responded at one month. Also, they were less frequently Turkish and were more likely to never have smoked than those who only responded at three months (S5 Table). No major differences in psychological outcomes and HRQoL were observed between full responders and partial responders, i.e., participants who only completed one follow-up assessment, besides full responders reporting less severe symptoms of anxiety and a higher overall HRQoL than those who only responded at 1 month (S6 Table). In addition, no differences were observed for the presence or severity of PTSD-, anxiety-, and depression-related symptomatology between responders and 12 randomly selected non-responders (S7 Table).

## Participants

Table 1 depicts the most relevant baseline demographics and treatment-related characteristics; a full overview of all collected demographics and characteristics can be found in S8 Table. The median age was 64 years (95% range 33–88), 106 (36%) were female, and a history of mental illness was reported by 27 of 201 patients (13%), who were willing to share this information. The median hospital length of stay (LOS) was 4 days (1–52) and 42 patients (14%) required ICU admission (median ICU LOS; 16 days [95% range 0–52]). Overall, 146 patients (50%) were SARS-CoV-2 positive. Non-COVID-19 patients were predominantly diagnosed with other pulmonary or cardiac illnesses (S8 Table). COVID-19 patients were younger, had fewer comorbidities, more frequently worked prior to hospitalization, had a longer hospital and ICU-LOS, were more frequently admitted to the ICU, had a higher SOFA score and lower P/F ratio at admission, more frequently received oxygen therapy and invasive ventilation for a longer duration, were more frequently mechanically ventilated in prone position, and more frequently received a tracheotomy than non-COVID-19 patients (Table 1). Within COVID-19 patients, those who were admitted to the ICU had a longer hospital LOS, a higher SOFA score and lower P/F ratio at admission, received oxygen therapy more frequently, and more frequently received a tracheostomy (Table 1).

## Post-traumatic stress, anxiety and depression

Table 2 depicts the descriptive statistics of psychological outcomes, and the comparisons between COVID-19 and non-COVID-19 patients, and between COVID-19 ICU and non-ICU admitted patients.

One month after hospital discharge, the median PTSD score was 9 (IES-R sum score, 95% range 0–48) and 16% of patients (39/237) reported probable PTSD. At three months, the PTSD score was decreased to 7 (0–44, $p = 0.01$), but the proportion of patients with probable PTSD was similar (28/209 [13%], $p = 0.06$). At both follow-up time-points, there were no differences in the severity of PTSD symptoms or the proportion of patients with probable PTSD between the COVID-19 and non-COVID-19 patients, nor between ICU and non-ICU patients.

One month after hospital discharge, 29% of patients (72/248) reported probable anxiety and the median anxiety score was 4 (HADS anxiety score, 95% range 0–17), which improved at three months after discharge to 20% of patients (42/208, $p = 0.01$) reporting probable anxiety and a median anxiety score of 4 (0–15, $p = 0.01$). At neither one of the follow-up time-points, differences were found in the severity of anxiety-related symptoms or the proportion of patients with probable anxiety between COVID-19 and non-COVID-19 patients or between COVID-19 ICU and non-ICU patients.

**Table 1. Baseline demographics and treatment-related characteristics of study participants.**

| | | Overall cohort | | | COVID-19 cohort | | |
|---|---|---|---|---|---|---|---|
| | **Overall** | **COVID-19** | **non-COVID-19** | **_p_-value** | **ICU** | **non-ICU** | **_p_-value** |
| Sample size | 294 | 146 | 148 | | 40 | 106 | |
| Age, years | 64 (33–88) | 61 (35–85) | 69 (31–89) | <0.001 | 62 (36–74) | 60 (33–86) | 0.37 |
| Ethnicity | | | | | | | |
| Caucasian | 207 (70%) | 92 (63%) | 115 (78%) | <0.01 | 22 (55%) | 70 (66%) | 0.25 |
| Black | 10 (3%) | 8 (5%) | 2 (1%) | 0.06 | 3 (8%) | 5 (5%) | 0.68 |
| Surinamese /Hindustan | 22 (7%) | 15 (10%) | 7 (5%) | 0.08 | 7 (18%) | 8 (8%) | 0.12 |
| Arab, not specified | 13 (4%) | 6 (4%) | 7 (5%) | 1.00 | 1 (3%) | 5 (5%) | 1.00 |
| Turkish | 8 (3%) | 5 (3%) | 3 (2%) | 0.50 | 0 (0%) | 5 (5%) | 0.32 |
| Moroccan | 10 (3%) | 8 (5%) | 2 (1%) | 0.06 | 4 (10%) | 4 (4%) | 0.22 |
| Others | 8 (3%) | 5 (3%) | 3 (2%) | 0.50 | 1 (3%) | 4 (4%) | 1.00 |
| Unknown | 16 (5%) | 7 (5) | 9 (6%) | 0.80 | 2 (5%) | 5 (5%) | 1.00 |
| Sex at birth, Female | 106 (36%) | 52 (36%) | 54 (36%) | 0.90 | 13 (33%) | 39 (37%) | 0.70 |
| Body Mass Index (BMI) * | 27.4 (19.3–43.2) | 28.0 (20.4–41.8) | 26.5 (18.9–43.1) | 0.30 | 28.5 (22.7–41.9) | 27.9 (19.7–39.5) | 0.35 |
| Total number of comorbidities | 1 (0–4) | 1 (0–3) | 2 (0–4) | <0.001 | 1 (0–3) | 1 (0–3) | 0.16 |
| Psychiatric problems in past 5 years ** | 27 (13%) | 12 (11%) | 15 (16%) | 0.41 | 3 (12%) | 9 (11%) | 1.00 |
| Educational level *** | | | | | | | |
| Elementary school | 46 (18%) | 21 (17%) | 25 (19%) | 0.75 | 4 (14%) | 17 (18%) | 0.78 |
| High school | 49 (20%) | 17 (14%) | 43 (25%) | 0.04 | 6 (21%) | 11 (12%) | 0.24 |
| Intermediate vocational education | 89 (40%) | 44 (36%) | 45 (35%) | 0.89 | 10 (34%) | 54 (37%) | 1.00 |
| Bachelor's degree | 40 (16%) | 20 (17%) | 20 (16%) | 0.86 | 4 (14%) | 16 (17%) | 0.78 |
| Master's degree | 26 (10%) | 19 (16%) | 7 (5%) | 0.01 | 5 (17%) | 14 (15%) | 0.78 |
| Working/employed before admission‡ | 92 (37%) | 59 (49%) | 33 (26%) | <0.001 | 12 (41%) | 47 (51%) | 0.40 |
| Healthcare worker | 18 (8%) | 13 (11%) | 5 (4%) | 0.08 | 3 (10%) | 10 (11%) | 1.00 |
| Cause of admission | | | | | | | |
| COVID-19 | 146 (50%) | 146 (100%) | 0 (0%) | N/A | 40 (100%) | 106 (100%) | N/A |
| Other | 148 (50%) | (0%) | 148 (50%) | N/A | (0%) | (0%) | N/A |
| Hospital length of stay, days | 4 (1–52) | 5 (1–60) | 4 (1–23) | <0.001 | 28 (9–69) | 4 (1–18) | <0.001 |
| Admitted to the ICU | 42 (14%) | 40 (27%) | 2 (1%) | <0.001 | 40 (100%) | 0 (0%) | N/A |
| ICU length of stay, days | 16 (0–52) | 16 (0–52) | 1 (1–2) | 0.01 | 16 (0–52) | N/A | N/A |
| SOFA score **** | 2 (0–6) | 2 (0–6) | 2 (0–6) | 0.04 | 3 (1–8) | 2 (0–5) | <0.001 |
| Received oxygen therapy | 222 (76%) | 135 (92%) | 87 (59%) | <0.001 | 40 (100%) | 95 (90%) | 0.04 |
| Received non-invasive ventilation (NIV) | 13 (4%) | 9 (6%) | 4 (3%) | 0.17 | 5 (13%) | 4 (4%) | 0.06 |
| Received invasive ventilation | 42 (14%) | 40 (27%) | 2 (1%) | <0.001 | 40 (100%) | 0 (0%) | N/A |
| Ventilated in prone position | 16 (5%) | 16 (11%) | 0 (0%) | <0.001 | 16 (40%) | N/A | N/A |
| Received a tracheostomy | 14 (5%) | 14 (10%) | 0 (0%) | <0.001 | 14 (35%) | 0 (0%) | <0.001 |
| Died during follow-up | 4 (1%) | 0 (0%) | 4 (3%) | 0.12 | 0 (0%) | 0 (0%) | 1.00 |

Data are shown as n (%) and median (95% range). Patients were stratified based on SARS-CoV2 PCR; COVID-19 and a non-COVID-19. COVID-19 patients were stratified based on necessity for intensive care treatment; COVID-19 ICU and COVID-19 non-ICU. Abbreviations: ICU, Intensive care unit; SOFA, Sequential Organ Failure Assessment; P/F ratio, ratio between arterial partial pressure (PaO2) to fractional inspired oxygen (FiO2); S/F ratio, ratio between peripheral oxygen saturation (SaO2) and FiO2. *P*-values were calculated using a Mann Whitney-U Test for continuous variables and using a Fisher's Exact test for categorical variables.

* BMI of 116 patients was not available.

** Results regarding psychological history are derived from the questionnaire 3 months after discharge. The proportions shown are calculated over a population of 212 patients.

*** Results regarding educational level are derived from the questionnaires 1 month after discharge. The proportions shown are calculated over a population of 252 patients.

**** Scored the day of first SARS-CoV-2 suspicion. Non-invasive ventilation was defined as use of CPAP or BIPAP; Use of high flow nasal cannula was not included.

**Table 2. Psychological outcomes throughout follow-up.**

| | | Overall cohort | | | | COVID-19 cohort | | | |
|---|---|---|---|---|---|---|---|---|---|
| | Overall | COVID-19 | non-COVID-19 | β / OR (95% CI) | *p*-value | ICU | non-ICU | β / OR (95% CI) | *p*-value |
| 1 month | n = 252 | 123 | 129 | N/A | N/A | 30 | 93 | N/A | N/A |
| 3 months | n = 212 | 116 | 96 | N/A | N/A | 33 | 83 | N/A | N/A |
| **PTSD** | | | | | | | | | |
| *Severity, median (95% range)* | | | | | | | | | |
| 1 month | 9 (0–48) | 9 (0–47) | 8 (0–48) | -0.8 (-4.2–2.7) | 0.67[1] | 7 (1–51) | 10 (0–46) | 0.9 (-5.0–6.7) | 0.77[10] |
| 3 months | 7 (0–44) | 7 (0–44) | 6 (0–38) | -1.0 (-4.1–2.2) | 0.55[2] | 8 (0–30) | 7 (0–45) | 0.9 (-4.2–6.0) | 0.73[10] |
| β, time (95% CI) | -2.8 (-3.1- -0.4) | -0.9 (-2.8–0.9) | -2.8 (-4.8- -0.8) | N/A | N/A | -2.3 (-4.6–0.1) | -0.6 (-2.9–1.8) | N/A | N/A |
| *p*-value (Time) | 0.01 | 0.32 | <0.001 | N/A | N/A | 0.08 | 0.64 | N/A | N/A |
| *Prevalence, n (%)* | | | | | | | | | |
| 1 month | 39 (16%) | 20 (17%) | 19 (16%) | 1.0 (0.5–2.2) | 0.95[2] | 5 (17%) | 15 (17%) | 1.1 (0.3–3.2) | 0.92[10] |
| 3 months | 28 (13%) | 17 (15%) | 11 (12%) | 0.8 (0.3–2.5) | 0.71[3] | 6 (18%) | 11 (13%) | 0.7 (0.2–2.0) | 0.50[10] |
| OR, time (95% CI) | 0.1 (0.0–1.1) | 0.3 (0.0–3.0) | 0.0 (0.0–4.4) | N/A | N/A | 0.0 (0.0–0.2) | 0.4 (0.0–5.0) | N/A | N/A |
| *p*-value (Time) | 0.06 | 0.29 | 0.15 | N/A | N/A | 0.02 | 0.47 | N/A | N/A |
| **Anxiety** | | | | | | | | | |
| *Severity, median (95% range)* | | | | | | | | | |
| 1 month | 4 (0–17) | 4 (0–15) | 5 (0–17) | 1.0 (-0.1–2.2) | 0.08[4] | 3 (0–15) | 4 (0–15) | 0.8 (-1.3–3.0) | 0.45[11] |
| 3 months | 4 (0–15) | 3 (0–15) | 4 (0–14) | 0.2 (-1.1–1.5) | 0.73[3] | 3 (0–15) | 4 (0–15) | 0.4 (-1.3–2.2) | 0.64[10] |
| β, time (95% CI) | -0.6 (-1.1- -0.1) | -0.4 (-1.1–0.2) | -0.8 (-1.5- -0.1) | N/A | N/A | -0.4 (-1.8–1.0) | -0.4 (-1.1–0.3) | N/A | N/A |
| *p*-value (Time) | 0.01 | 0.19 | 0.02 | N/A | N/A | 0.61 | 0.22 | N/A | N/A |
| *Prevalence, n (%)* | | | | | | | | | |
| 1 month | 72 (29%) | 30 (25%) | 42 (33%) | 1.3 (0.6–2.4) | 0.50[5] | 5 (17%) | 25 (27%) | 2.4 (0.7–8.0) | 0.15[11] |
| 3 months | 42 (20%) | 20 (17%) | 22 (24%) | 1.4 (0.6–3.4) | 0.42[3] | 5 (15%) | 15 (18%) | 0.8 (0.3–2.4) | 0.69[10] |
| OR, time (95% CI) | 0.2 (0.1–0.7) | 0.2 (0.0–1.1) | 0.2 (0.0–1.3) | N/A | N/A | 0.2 (0.0–10.7) | 0.6 (0.3–1.2) | N/A | N/A |
| *p*-value (Time) | 0.01 | 0.06 | 0.09 | N/A | N/A | 0.46 | 0.16 | N/A | N/A |
| **Depression** | | | | | | | | | |
| *Severity, median (95% range)* | | | | | | | | | |
| 1 month | 5 (0–16) | 3 (0–16) | 6 (0–17) | 0.9 (-0.4–2.1) | 0.17[6] | 4 (0–14) | 3 (0–16) | 0.8 (-1.2–2.8) | 0.44[10] |
| 3 months | 4 (0–16) | 3 (0–13) | 4 (0–17) | -0.4 (-1.8–1.0) | 0.54[7] | 3 (0–12) | 3 (0–13) | 1.1 (-0.5–2.8) | 0.19[10] |
| β, time (95% CI) | -0.6 (-1.1- -0.1) | -0.3 (-0.9–0.3) | -0.9 (-1.7- -0.2) | N/A | N/A | -0.4 (-1.1–0.3) | -0.2 (-1.0–0.6) | N/A | N/A |
| *p*-value (Time) | 0.02 | 0.35 | 0.02 | N/A | N/A | 0.27 | 0.57 | N/A | N/A |
| *Prevalence, n (%)* | | | | | | | | | |
| 1 month | 79 (32%) | 32 (26%) | 47 (37%) | 1.2 (0.7–2.2) | 0.55[8] | 3 (10%) | 29 (32%) | 4.2 (1.2–15.0) | 0.03[10] |
| 3 months | 50 (24%) | 25 (22%) | 25 (27%) | 0.6 (0.2–1.3) | 0.18[9] | 2 (6%) | 23 (28%) | 6.0 (1.3–27.3) | 0.02[10] |
| OR, time (95% CI) | 0.6 (0.3–1.0) | 0.4 (0.1–1.8) | 0.5 (0.2–1.1) | N/A | N/A | 0.6 (0.1–3.7) | 0.8 (0.4–1.6) | N/A | N/A |
| *p*-value (Time) | 0.06 | 0.23 | 0.10 | N/A | N/A | 0.57 | 0.58 | N/A | N/A |

Descriptive statistics of the psychological distress outcomes, stratified by COVID-19 diagnosis and ICU admission. Severity of PTSD, anxiety, and depression were expressed as the IES-R, HADS anxiety, and HADS depression sum scores, respectively. Prevalence of probable PTSD, anxiety, and depression was defined as the proportion of patients scoring above the cut-off. Abbreviations: CI, confidence interval; COVID-19, coronavirus disease 2019; ICU, intensive care unit; OR, odds ratio; PTSD, post-traumatic stress disorder. Differences over time were calculated using mixed effects linear (for continuous outcomes) and logistic (for categorical outcomes) regression models, with time as independent variable. Differences between stratifications were analyzed using simple linear (for continuous outcomes) and logistic (for categorical outcomes) regression models.

[1] Adjusted for ethnicity and educational level

[2] adjusted for ethnicity

[3] Adjusted for ethnicity and employment status before hospitalization

[4] adjusted for ethnicity, educational level, and SOFA score during admission

[5] adjusted for ethnicity, educational level, ICU admission, and SOFA score during admission

[6] adjusted for ethnicity, educational level, and ICU admission

[7] adjusted for age, ethnicity, educational level, and employment status before hospitalization

[8] adjusted for ethnicity, ICU admission, and employment status before hospitalization

[9] adjusted for age, educational level, ICU admission, and employment status before hospitalization

[10] not adjusted

[11] adjusted for SOFA score during admission.

The median depression score was 5 (95% range 0–16) and 32% of patients (79/248) reported probable depression at one month post-discharge. While the severity of depression-related symptoms improved at three months to a median of 4 (95% range 0–16, $p = 0.02$), there was no difference between the proportion of patients reporting probable depression. No differences were found in the severity of depression between COVID-19 and non-COVID-19 patients, nor between COVID-19 ICU and non-ICU patients. Although no differences were observed in the proportion of patients reporting probable depression between COVID-19 and non-COVID-19 patients, within the COVID-19 cohort, ICU patients less frequently reported probable depression (1 month, OR 4.2 [95% CI 1.2–15.0], $p = 0.03$; 3 months, OR 6.0 [95% CI 1.3–27.3], $p = 0.02$; Table 2).

## Health-related quality of life

Table 3 depicts the descriptive statistics of HRQoL outcomes, and the comparisons between COVID-19 and non-COVID-19 patients and between COVID-19 ICU and non-ICU patients.

The median overall HRQoL score was 0.74 (EQ5D TTO score, 95% range -0.08–1.00) and the self-reported health was 65 (EQ5D VAS score, 10–95) at one month post-discharge. Both improved three months after discharge (EQ5D TTO: estimated mean difference over time = 0.04 [95% CI 0.01–0.07], $p = 0.01$; EQ5D VAS: estimated mean difference over time = 4.19 [95% CI 1.03–7.35], $p = 0.01$). There were no differences in HRQoL or the self-reported health between COVID-19 and non-COVID-19 patients or between COVID-19 ICU and non-ICU patients.

The median mental quality of life was 39 (median MSC-36, 96% range 15–63) at one month after hospital discharge and improved to 48 (19–64, $p<0.001$) at three months post-discharge. The median physical quality of life was 37 (median PCS-36, 18–56) and improved to 39 (17–59, $p<0.01$). COVID-19 patients reported a better physical quality of life at 1 month (estimated mean difference -3.3 [95% CI -6.1 - -0.5, $p = 0.02$; Table 3), but not at three months, and no differences in mental quality of life were observed. COVID-19 ICU patients had a better mental quality of life at 3 months (estimated mean difference -7.6 [95% CI -13.5 - -1.8], $p = 0.01$; Table 3), but not at 1 month, and no differences were observed in the physical quality of life within the COVID-19 cohort between ICU and non-ICU patients.

## Predictors of psychological distress and quality of life

Tables 4 and S9 depict the results of the exploration of predictors for the severity and prevalence of probable PTSD, anxiety, and depression. Time after discharge ($p = 0.01$), not being Caucasian ($p<0.01$) and having completed higher professional education were associated with the severity of PTSD in the univariate mixed effects regression analyses and were included in the multivariate mixed effects regression analysis. Of these, time after discharge (estimated mean difference (β) = 1.8 [95% CI -3.2 - -0.4], $p = 0.01$) negatively and not being Caucasian (β = 5.2 [95% CI 1.8–8.6], $p = 0.003$; Table 4), positively predicted the severity of PTSD. None of the predictors were significantly associated with the prevalence of probable PTSD (S9 Table).

Time after discharge ($p = 0.01$), female gender ($p = 0.01$), having completed higher professional education ($p = 0.01$), employment status before hospitalization ($p = 0.08$), and COVID-19 diagnosis ($p = 0.06$) were associated with the severity of anxiety in the univariate mixed effects regression analyses and were included in the multivariate mixed effects regression analysis. Of these, time after discharge (β = -0.6 [95% CI -1.1 - -0.1], $p = 0.02$) negatively and not being Caucasian (1.9 [95% CI 0.6–3.1], $p = 0.01$) positively predicted the severity of anxiety (Table 4). None of the predictors were significantly associated with the prevalence of probable anxiety (S9 Table).

**Table 3. Health-related quality of life throughout follow-up.**

| | | Overall cohort | | | | COVID-19 cohort | | | |
|---|---|---|---|---|---|---|---|---|---|
| | Overall | COVID-19 | non-COVID-19 | β/OR (95% CI) | *p*-value | ICU | non-ICU | β/OR (95% CI) | *p*-value |
| 1 month | n = 252 | 123 | 129 | N/A | N/A | 30 | 93 | N/A | N/A |
| 3 months | n = 212 | 116 | 96 | N/A | N/A | 33 | 83 | N/A | N/A |
| **Overall HRQoL, *median (95% range)*** | | | | | | | | | |
| 1 month | 0.7 (-0.1–1.0) | 0.8 (0.0–1.0) | 0.7 (-0.2–1.0) | -0.1 (-0.1–0.01) | 0.10[1] | 0.7 (0.0–1.0) | 0.8 (0.0–1.0) | 0.03 (-0.1–0.2) | 0.52[2] |
| 3 months | 0.8 (0.1–1.0) | 0.8 (0.1–1.0) | 0.7 (0.0–1.0) | 0.0 (-0.1–0.04) | 0.37[1] | 0.8 (0.1–1.0) | 0.8 (0.1–1.0) | 0.04 (-0.1–0.1) | 0.32[2] |
| β, time (95% CI) | 0.0 (0.0–0.1) | 0.0 (0.0–0.1) | 0.0 (0.0–0.1) | N/A | N/A | 0.0 (-0.1–0.1) | 0.05 (0.0–0.1) | N/A | N/A |
| *p*-value (Time) | 0.01 | 0.07 | 0.11 | N/A | N/A | 0.98 | 0.04 | N/A | N/A |
| **Perceived Health State, *median (95% range)*** | | | | | | | | | |
| 1 month | 65 (10–95) | 70 (25–95) | 60 (10–90) | -5.1 (-10.8–0.6) | 0.08[3] | 68 (9–95) | 70 (30–99) | 6.1 (-2.0–14.1) | 0.14[2] |
| 3 months | 73 (9–95) | 75 (18–95) | 70 (8–95) | -2.8 (-3.9–9.5) | 0.41[4] | 70 (8–96) | 75 (20–95) | 4.4 (-4.1–12.9) | 0.31[2] |
| β, time (95% CI) | 4.2 (1.1–7.4) | 3.4 (-0.2–7.1) | 5.0 (-0.4–10.4) | N/A | N/A | 4.1 (-4.1–12.3) | 3.2 (-0.8–7.3) | N/A | N/A |
| *p*-value (Time) | 0.01 | 0.07 | 0.07 | N/A | N/A | 0.33 | 0.13 | N/A | N/A |
| **Mental HRQoL, *median (95% range)*** | | | | | | | | | |
| 1 month | 39 (15–63) | 40 (14–62) | 39 (16–63) | 2.7 (-0.8–6.1) | 0.13[5] | 46 (17–62) | 39 (13–60) | -5.1 (-10.8–0.5) | 0.08[2] |
| 3 months | 48 (19–64) | 49 (21–64) | 47 (19–64) | 2.6 (-1.6–6.7) | 0.23[6] | 53 (29–64) | 48 (19–62) | -7.6 (-14 - -1.8) | 0.01[7] |
| β, time (95% CI) | 6.0 (4.3–7.7) | 7.2 (4.8–9.7) | 4.6 (2.2–7.0) | N/A | N/A | 7.5 (3.5–11.5) | 7.1 (4.1–10.1) | N/A | N/A |
| *p*-value (Time) | <0.001 | <0.001 | <0.001 | N/A | N/A | 0.001 | <0.001 | N/A | N/A |
| **Physical HRQoL, *median (95% range)*** | | | | | | | | | |
| 1 month | 37 (18–56) | 39 (22–56) | 35 (15–56) | -3.3 (-6.1 - -0.5) | 0.02[8] | 38 (23–57) | 39 (22–56) | -4.1 (-10.3–2.0) | 0.18[10] |
| 3 months | 39 (17–59) | 41 (22–58) | 37 (16–59) | -2.2 (-6.0–1.6) | 0.26[9] | 40 (19–58) | 42 (22–59) | 1.2 (-3.5–5.9) | 0.63[2] |
| OR, time (95% CI) | 1.7 (0.5–3.0) | 2.4 (0.6–4.1) | 0.8 (-0.8–2.4) | N/A | N/A | 3.0 (-0.7–6.7) | 2.2 (0.2–4.2) | N/A | N/A |
| *p*-value (Time) | <0.01 | 0.01 | 0.35 | N/A | N/A | 0.12 | 0.03 | N/A | N/A |

Descriptive statistics of the HRQoL outcomes, stratified by COVID-19 diagnosis and ICU admission. Differences over time were calculated using mixed effects linear (for continuous outcomes) and logistic (for categorical outcomes) regression models, with time as independent variable. Abbreviations: CI, confidence interval; COVID-19, coronavirus disease 2019; EQ-5D, European quality of life 5 dimensions questionnaire; HRQoL, health-related quality of life; ICU, intensive care unit; MCS-36, mental component score of the RAND-36; OR, odds ratio; PCS-36, physical component score of the RAND-36; TTO, time trade-off; VAS, visual analogue scale. Differences between stratifications were analyzed using simple linear (for continuous outcomes) and logistic (for categorical outcomes) regression models.

[1] Adjusted for age, ethnicity, educational level, and employment status before hospitalization

[2] not adjusted

[3] adjusted for age, ethnicity and employment status before hospitalization

[4] adjusted for ethnicity and employment status before hospitalization

[5] adjusted for ethnicity and educational level

[6] adjusted for ethnicity, ICU admission, employment status before hospitalization and SOFA score during admission

[7] adjusted SOFA score during admission

[8] adjusted for age, ethnicity, hospital length of stay, and employment status before hospitalization

[9] adjusted for age, educational level, and employment status before hospitalization

[10] not hospital length of stay.

Time after discharge (*p* = 0.01), female gender (*p* = 0.02), not being Caucasian (*p* = 0.06), having completed higher professional (*p*<0.01) or university education (*p* = 0.01), employment status before hospitalization (*p* = 0.02), COVID-19 diagnosis (*p* = 0.03) and ICU admission (*p* = 0.07) were associated with the severity of depression in the univariate mixed effects regression analyses and were included in the multivariate mixed effects regression analysis. Of

**Table 4. Predictors for the severity of PTSD, anxiety, and depression.**

| | Severity of PTSD | | | | Severity of anxiety | | | | Severity of depression | | | |
|---|---|---|---|---|---|---|---|---|---|---|---|---|
| | *Univariate* | | *Multivariate* | | *Univariate* | | *Multivariate* | | *Univariate* | | *Multivariate* | |
| | **Beta (95% CI)** | **P** | **Beta** | **P** | **Beta** | **P** | **Beta** | **P** | **Beta** | **P** | **Beta** | **P** |
| Time *(3 months)* | -1.8 (-3.1 - -0.4) | **0.01** | -1.8 (-3.2 - -0.4) | **0.01** | -0.6 (-1.1 - -0.2) | **0.01** | -0.6 (-1.1 - -0.1) | **0.02** | -0.6 (-1.1 - -0.1) | **0.01** | -0.6 (-1.1 - -0.1) | **0.03** |
| Age, *years* | -0.1 (-0.2– 0.04) | 0.29 | | | 0.0 (-0.1– 0.01) | 0.10 | | | 0.0 (-0.01– 0.1) | 0.90 | | |
| Gender, *(female)* | 2.3 (-0.8–5.3) | 0.14 | | | 1.4 (0.3–2.5) | **0.01** | 1.1 (-0.1–2.2) | 0.06 | 1.3 (0.2–2.3) | **0.02** | 0.7 (-0.4–1.9) | 0.21 |
| Ethnicity *(Non-Caucasian)* | 6.2 (3.1–9.3) | **<0.01** | 5.2 (1.8–8.6) | **<0.01** | 1.8 (0.6–2.9) | **<0.01** | 1.9 (0.6–3.1) | **<0.01** | 1.1 (0.0–2.3) | 0.06 | 1.1 (-0.1–2.3) | 0.08 |
| Educational level | | | | | | | | | | | | |
| *(High school)* | 2.8 (-2.3–7.9) | 0.28 | 3.5 (-1.4–8.5) | 0.16 | 0.4 (-1.4–2.2) | 0.67 | 0.3 (-1.5–2.1) | 0.74 | 0.2 (-1.5–2.0) | 0.80 | 0.2 (-1.5–2.0) | 0.81 |
| *(Vocational)* | 1.0 (-3.4–5.5) | 0.65 | 2.3 (-2.1–6.7) | 0.30 | 0.1 (-1.5–1.7) | 0.86 | 0.8 (-0.8–2.4) | 0.34 | -0.4 (-1.9–1.2) | 0.66 | 0.2 (-1.4–1.9) | 0.76 |
| *(Higher professional)* | -4.7 (-9.9–0.5) | **0.08** | -3.3 (-8.5–1.9) | 0.22 | -2.7 (-4.6 - -0.7) | **0.01** | -1.8 (-3.8–0.1) | 0.06 | -2.6 (-4.5 - -0.8) | **<0.01** | -1.9 (-3.8–0.0) | **0.05** |
| *(University)* | -4.4 (-10.3–1.4) | 0.14 | -3.0 (-8.7–2.8) | 0.31 | -1.7 (-3.8–0.5) | 0.13 | -0.8 (-2.9–1.4) | 0.49 | -2.7 (-4.1 - -0.7) | **0.01** | -1.8 (-3.9–0.3) | 0.09 |
| Employed *(Yes)* | -2.2 (-5.4–1.0) | 0.18 | | | -1.1 (-2.2–0.1) | **0.08** | -0.7 (-1.9–0.5) | 0.28 | -1.4 (-2.5 - -0.3) | **0.02** | -0.9 (-2.1–0.3) | 0.14 |
| Healthcare worker *(Yes)* | 4.9 (-1.2–10.9) | 0.12 | | | -0.2 (-2.4–2.0) | 0.88 | | | 0.3 (-1.9–2.5) | 0.81 | | |
| Hospital LOS, days | 0.1 (-0.04–0.2) | 0.22 | | | -0.01 (-0.1–0.0) | 0.48 | | | 0.0 (-0.1–0.03) | 0.63 | | |
| Mechanical ventilation *(Yes)* | 1.4 (-3.0–5.7) | 0.54 | | | -0.4 (-2.0–1.2) | 0.61 | | | -1.0 (-2.6–0.6) | 0.21 | | |
| SOFA admission score | 0.0 (-0.1–0.1) | 0.99 | | | -0.2 (-0.5–0.1) | 0.25 | | | -0.1 (-0.5–0.2) | 0.37 | | |
| COVID-19 *(Yes)* | 0.9 (-2.0–3.8) | 0.52 | | | -1.0 (-2.1–0.0) | **0.06** | -0.9 (-2.0–0.2) | 0.12 | -1.2 (-2.2 - -0.1) | **0.03** | -0.4 (-1.6–0.8) | 0.48 |
| ICU admission *(Yes)* | 0.7 (-3.4–4.8) | 0.74 | | | -0.7 (-2.2–0.7) | 0.33 | | | -1.4 (-2.9–0.1) | **0.07** | -1.6 (-3.3–0.1) | 0.07 |

Univariate analysis was performed using mixed effects linear regression models, with the variable of interest as independent variable, and a random intercept for each participant. Each variable with a p-value <0.10 in the univariate mixed model was implemented as independent variable in the multivariate mixed model. Variables with a p-value <0.05 in the multivariate mixed model were identified as independent predictors.

these, time after discharge positively ($\beta$ = -0.6 [95% CI -1.1 - -0.1], $p$ = 0.03) and completed higher professional education ($\beta$ = -1.9 [-3.8–0.0], $p$ = 0.049) negatively predicted the severity of depression (Table 4). None of the predictors were significantly associated with the prevalence of probable depression (S9 Table).

Table 5 depict the results of the exploration of predictors for the overall, mental, and physical HRQoL. Time after discharge ($p$ = 0.01), age ($p$<0.01), female gender ($p$<0.01), having completed higher professional ($p$<0.01) or university education ($p$<0.01) and COVID-19 diagnosis ($p$<0.01) were associated with overall HRQoL in the univariate mixed effects regression analyses and were included in the multivariate mixed effects regression analysis. Of these, having completed university education ($\beta$ = 0.11 [95% CI 0.002–0.21], $p$ = 0.046) and being employed ($\beta$ = 0.12 [95% CI 0.06–0.19], $p$<0.01) positively, and having more severe symptoms of anxiety ($\beta$ = -0.03 [95% CI -0.04 - -0.02], $p$<0.001) or depression at 1 month ($\beta$ = -0.01 [95% CI -0.02 - -0.001], $p$ = 0.04; Table 5) negatively predicted the overall HRQoL. The exploration of predictors of the perceived health state is depicted in S9 Table.

**Table 5. Predictors for the overall HRQol, the mental quality of life, and the physical quality of life.**

| | Overall HRQoL | | | | Mental HRQoL | | | | Physical HRQoL | | | |
|---|---|---|---|---|---|---|---|---|---|---|---|---|
| | Univariate | | Multivariate | | Univariate | | Multivariate | | Univariate | | Multivariate | |
| | Beta (95% CI) | P | Beta | P | Beta | P | Beta | P | Beta | P | Beta | P |
| Time (3 months) | 0.03 (0.01–0.07) | **0.01** | 0.03 (-0.003–0.06) | 0.08 | 6.1 (4.3–7.8) | **<0.01** | 5.6 (3.9–7.4) | **<0.01** | 1.8 (0.6–3.0) | **<0.01** | 1.6 (0.3–2.8) | **0.02** |
| Age, years | -0.00 (-0.01–0.00) | **<0.01** | -0.001 (-0.003–0.002) | 0.58 | 0.1 (-0.1–0.2) | 0.31 | | | -0.2 (-0.3 - -0.1) | **<0.01** | -0.1 (-0.2–0.02) | 0.11 |
| Gender (female) | -0.11 (-0.18 - -0.04) | **<0.01** | -0.03 (-0.08–0.03) | 0.35 | -4.1 (-7.1 - -1.1) | **<0.01** | -1.9 (-4.3–0.5) | 0.13 | -4.5 (-7.1 - -1.9) | **<0.01** | -4.1 (-6.6 - -1.6) | **<0.01** |
| Ethnicity (Non-Caucasian) | -0.01 (-0.06–0.08) | 0.70 | | | -3.2 (-6.4–0.0) | **0.05** | 0.03 (-2.6–2.7) | 0.98 | 2.0 (-0.7–4.8) | 0.15 | | |
| Educational level | | | | | | | | | | | | |
| (High school) | 0.00 (-0.11–0.12) | 0.94 | 0.03 (-0.05–0.12) | 0.49 | -0.1 (-5.2–5.0) | 0.96 | 1.2 (-2.6–5.1) | 0.53 | -2.9 (-7.1–1.4) | 0.20 | | |
| (Vocational) | 0.07 (-0.03–0.17) | 0.16 | 0.01 (-0.07–0.09) | 0.81 | -1.1 (-5.5–3.4) | 0.65 | -1.5 (-4.9–2.0) | 0.40 | -1.0 (-4.7–2.8) | 0.60 | | |
| (Higher professional) | 0.20 (0.08–0.32) | **<0.01** | 0.05 (-0.05–0.15) | 0.33 | 4.1 (-1.2–9.4) | 0.13 | -0.4 (-4.4–3.6) | 0.84 | 3.0 (-1.4–7.4) | 0.19 | | |
| (University) | 0.23 (0.14–0.41) | **<0.01** | 0.11 (0.002–0.21) | **0.05** | 6.5 (0.6–12.3) | **0.03** | 2.1 (-2.2–6.4) | 0.34 | 3.8 (-1.1–8.8) | 0.13 | | |
| Employed (Yes) | 0.17 (0.10–0.25) | **<0.01** | 0.12 (0.06–0.19) | **<0.01** | 1.5 (-1.7–4.7) | 0.35 | | | 4.6 (2.0–7.2) | **<0.01** | 1.9 (-0.9–4.7) | 0.19 |
| Healthcare worker (Yes) | 0.11 (-0.03–0.25) | 0.15 | | | -3.5 (-9.8–2.8) | 0.28 | | | -1.6 (-7.0–3.8) | 0.56 | | |
| Hospital LOS, days | 0.00 (0.00–0.00) | 0.59 | | | 0.1 (-0.1–0.2) | 0.33 | | | -0.1 (-0.2–0.01) | **0.07** | -0.2 (-0.3 - -0.1) | **<0.01** |
| Mechanical ventilation (Yes) | 0.02 (-0.08–0.13) | 0.67 | | | 5.0 (0.6–9.4) | **0.03** | -0.04 (-7.9–7.9) | 0.99 | -0.3 (-4.1–3.6) | 0.90 | | |
| SOFA admission score | 0.00 (-0.02–0.02) | 0.72 | | | 0.5 (-0.4–1.3) | 0.31 | | | -0.3 (-1.0–0.5) | 0.44 | | |
| COVID-19 (Yes) | 0.11 (0.04–0.18) | **<0.01** | 0.03 (-0.03–0.08) | 0.38 | 0.3 (-2.6–3.2) | 0.84 | | | 3.7 (1.2–6.2) | **<0.01** | 2.9 (0.3–5.5) | **0.03** |
| ICU admission (Yes) | 0.04 (-0.05–0.14) | 0.43 | | | 5.3 (1.2–9.4) | **0.01** | 0.9 (-6.5–8.3) | 0.81 | 0.7 (-2.9–4.2) | 0.71 | | |
| PTSD severity at 1 month | -0.097 (-0.010 - -0.005) | **<0.01** | 0.002 (-0.001–0.004) | 0.22 | -0.5 (-0.6 - -0.4) | **<0.01** | -0.1 (-0.2–0.04) | 0.20 | -0.2 (-0.3 - -0.1) | **<0.01** | 0.0 (-0.1–0.1) | 0.93 |
| Anxiety severity at 1 month | -0.04 (-0.04 - -0.03) | **<0.01** | -0.03 (-0.04 - -0.02) | **<0.01** | -1.8 (-2.1 - -1.6) | **<0.01** | -0.8 (-1.2 - -0.3) | **<0.01** | -0.8 (-1.0 - -0.5) | **<0.01** | -0.2 (-0.7–0.2) | 0.37 |
| Depression severity at 1 month | -0.04 (-0.04 - -0.03) | **<0.01** | -0.01 (-0.02 - -0.001) | **0.04** | -1.8 (-2.1 - -1.6) | **<0.01** | -1.0 (-1.4 - -0.6) | **<0.01** | -0.9 (-1.2 - -0.6) | **<0.01** | -0.6 (-1.0 - -0.2) | **<0.01** |

Univariate analysis was performed using mixed effects linear regression models, with the variable of interest as independent variable, and a random intercept for each participant. Each variable with a p-value <0.10 in the univariate mixed model was implemented as independent variable in the multivariate mixed model. Variables with a p-value <0.05 in the multivariate mixed model were identified as independent predictors.

Time after discharge ($p<0..01$), female gender ($p<0.01$), being not Caucasian ($p = 0.05$), having completed university education ($p = 0.03$), being mechanically ventilated ($p = 0.03$), ICU admission ($p = 0.01$) and having more severe symptoms of PTSD ($p<0.01$), anxiety ($p<0.01$) and depression ($p<0.01$) were associated with mental HRQoL in the univariate mixed effects regression analyses and were included in the multivariate mixed effects regression analysis. Of these, time after discharge (β = 5.6 [95% CI 3.9–7.4], $p<0.01$) positively, and having more severe symptoms of anxiety (β = -0.8 [95% CI -1.2 - -0.3], $p<0.01$) or depression (β = -1.0 [95% CI -1.4 - -0.6], $p<0.01$) at 1 month negatively predicted the mental HRQoL.

Lastly, time after discharge ($p<0.01$), age ($p<0.01$), female gender ($p<0.01$), employment status before hospitalization ($p<0.01$), hospital LOS ($p = 0.07$), COVID-19 diagnosis ($p<0.01$) and having more severe symptoms of PTSD ($p<0.01$), anxiety ($p<0.01$) and depression ($p<0.01$) were associated with physical HRQoL in the univariate mixed effects regression analyses and were included in the multivariate mixed effects regression analysis. Of these, time after discharge ($\beta$ = 1.6 [95% CI 0.3–2.8], $p = 0.02$) and COVID-19 diagnosis ($\beta$ = 2.9 [95% CI 0.3–5.5], $p = 0.03$] positively, and female gender ($\beta$ = -4.1 [95% CI -6.6 - -1.6], $p<0.01$), hospital LOS ($\beta$ = -0.2 [95% CI -0.3 - -0.1], $p<0.01$) and having more severe symptoms of depression ($\beta$ = -0.6 [95% CI -1.0 - -0.2], $p<0.01$) at 1 month negatively predicted the physical HRQoL.

## Discussion

In this observational study, assessing psychological distress and HRQoL in patients hospitalized during the COVID-19 pandemic, we observed that a substantial proportion of patients reported psychological distress, but no differences were observed in its severity and prevalence between patients with or without COVID-19. COVID-19 ICU patients suffered less from depression than their non-ICU counterparts. Post-discharge HRQoL was poor in all patients but improved during follow-up.

The proportion of patients with psychological distress as found in our study at one month post-discharge is comparable with proportions observed in recent studies, but data on psychological recovery beyond two months remains scarce [23,24,39]. Early psychological evaluation is disputable as for example PTSD can only be diagnosed as PTSD at one month after the traumatic event, and some level of depression and/or anxiety is not considered pathological in the first weeks post-discharge and may subside with time [40]. To distinguish between psychological impairments and a normal distress response to hospitalization, follow-up beyond one month is mandatory. Although the number of patients with psychological symptoms in our study decreased over time, we observed that psychological distress at three months remains substantial, more reliable illustrating the true post-discharge burden of hospitalization during the pandemic. Furthermore, extended follow-up up beyond 3 months will enable further confirmation of this burden and its trajectory over time.

The observed prevalence of psychological distress in our cohort appears to be higher than in the general population during non-pandemic circumstances, and was similar to people suffering from PTSD-related symptoms after a traumatic event, such as theft, burglary, accidents and death of a significant other [41]. The underlying cause of psychological distress in our population is most likely multifactorial and may include policies surrounding the pandemic containment, experiencing traumatic events, such as hospitalization and isolation, and the primary illness causing hospitalization. During previous coronavirus epidemics and in the wake of the COVID-19 pandemic, general population's mental health appeared to be reduced [25,42,43]. In-hospital universal isolation measures, such as implemented during a pandemic, have also shown to increase the risk for anxiety and depression [44]. Approximately 12% of general hospitalized patients suffer from depression, a percentage that like most stress-related psychological disorders is expected to decline over time [7,9]. Moreover, post-discharge depression, anxiety, and PTSD are more prevalent in hospitalized patients with a specific history of traumatizing events, such as cardiac arrest or unintentional injury [2,4,5,8]. Severe illnesses, such as sepsis, septic shock, and acute respiratory distress syndrome, increase the risk of psychological disorders [10,45]. COVID-19 itself may also be a contributing factor. Already early during the pandemic, concerns were raised that especially COVID-19 patients were at substantial risk for psychological morbidity based on previous experiences with similar coronavirus epidemics and possible neurotropic effects of SARS-CoV-2 [25,46]. In a recent large

retrospective study, COVID-19 was indeed associated with a higher risk of psychiatric outcomes as compared to influenza and respiratory tract infections [29]. In our cohort however, psychological distress was largely similar between non-COVID-19 patients with respiratory symptoms and COVID-19 patients, and we did not observe COVID-19 positivity to be a predictor for severity of PTSD, anxiety, or depression. As such, our data collectively suggest that COVID-19 on its own has no major influence on psychological outcome. The contradictory findings of the Taquet and our study will largely be explained by the difference in study design (retrospective cohort vs. prospective observational cohort), the different outcome measures used (ICD-based diagnoses vs. questionnaire-based outcome) and comparing the covid-19 cohort with other control populations (influenza and respiratory tract infection vs. patient with symptoms suggestive of COVID-19 but COVID-19 PCR negatives). Several considerations regarding the interpretation of our findings should however be taken into account. First, non-COVID-19 patients suffered from more comorbidities and were older, indicative of a worse pre-existent health status. In our cohort, psychological status and HRQoL prior to hospitalization were not available and thus we cannot formally rule out that pre-existing psychological well-being was poorer, predisposing for impaired psychological recovery [47]. Moreover, a poorer pre-hospitalization health may also explain the lower HRQoL post-discharge in non-COVID-19 patients. Secondly, national and regional initiatives resulted in extensive aftercare programs for COVID-19 patients, including psychological assessment, and referral to a psychologist when necessary [48,49]. Although data on the effectiveness of such post-COVID programs are lacking, similar programs were scarcely available for non-COVID-19 patients and only started 6 weeks post-discharge, so we cannot exclude some underestimation of the psychological burden in our COVID-19 population at three months.

Irrespective of the role of SARS-CoV-2 in the development of psychological distress, our data argue against an almost exclusive focus on COVID-19 patients for in-hospital and post-hospital care to improve recovery. A more appropriate strategy could be to aim at those at highest risk. Previous studies pinpointed ICU patients as those at high risk for psychological sequelae [10,45,50]. Although ICU admission was not predictive for severity of psychological distress, several other predictors identified in our cohort, i.e., ethnicity and educational level, are consistent with previously described risk factors for post-ICU psychological trauma [51]. Of note, the proportion of post-ICU patients suffering from psychological morbidity in our study was considerably lower than previously reported in both general as COVID-19 ICU survivors [12,15,21,45,52–55]. Depression was even less prevalent in ICU COVID-19 patients compared to their non-ICU counterparts. We can only speculate on possible explanations as literature on psychological recovery of COVID-19 ICU patients is currently scarce. A possible explanation may be that most ICU patients required deep sedation to facilitate lung protective ventilation [56]. As a result, patients may have experienced fewer anxious or delusional memories and were less aware of the severity of their illness, which are important contributors for psychological impairments following ICU discharge [57]. In contrast to ICU patients, non-ICU patients were fully awake during their hospital stay, and aware of the severity of their illness, potentially causing higher stress levels [57]. Also, ICU patients were, in general more frequently male and younger than ICU patients in a non-COVID-19 setting, and most often had no history of mental illness, possibly making them less susceptible for developing psychologic distress [51,58,59].

## Limitations

Several limitations of our study should be acknowledged. First, the high incidence of psychological symptomatology and the poor HRQoL post-discharge may be either attributed to

factors related to hospitalization or to baseline psychological imbalances, as psychological status and HRQoL of participants prior to hospitalization was not available. However, only a few reported a history of mental illness, suggesting that hospitalization rather than pre-existing psychological distress relates to post-discharge symptomatology. To overcome this issue, we believe that future studies on psychological and HRQoL recovery should strive for a best effort to obtain a pre-exposure evaluation [60]. Second, the single-center design limits its external validity. In-hospital measures taken to minimize further spread of the virus and treatment protocols were however comparable to those in hospitals in the Netherlands and other high-income countries [49]. Additionally, demographic and treatment-related characteristics of our COVID-19 patients are in line with previously described COVID-19 cohorts [61]. Third, although the overall sample size was substantial, our study was only sufficiently powered to detect major differences between patient subpopulations, limiting elaborate assessment of predictors for psychological distress. Ongoing larger follow-up studies of specific populations are warranted to extend these exploratory observations. Fourth, we assessed psychological well-being and HRQoL using self-report questionnaires. Although commonly used and extensively validated, formal assessment of psychologic disorders requires consultation with a psychologist or psychiatrist, and usage of self-reports may result in an overestimation of psychologic distress. Lastly, the number of non-responders in our study was relatively high, a problem common in the field of longitudinal follow-up research [62]. Baseline characteristics of responders were however comparable to non-responders, and in a sample of non-responders' psychological distress was roughly similar to responders. Collectively, our cohort was comparable with the overall population at baseline and for psychological outcomes. Baseline characteristics (demographic and treatment-related factors) were similar for non-responders, responders, and a sample of non-responders willing to respond once. In addition, psychological outcomes and HRQoL were comparable between responders and the sample of non-responders willing to respond, and between full and partial responders.

## Conclusions

In conclusion, more than one-third of patients admitted during the COVID-19 pandemic suffers from PTSD, anxiety, depression, or a combination, and a poor HRQoL. Physicians should be aware of the psychological consequences of hospitalization during a pandemic, and that psychological distress not only occurs in those affected by COVID-19 or those requiring ICU treatment. Appropriate psychological support and aftercare may be equally important to improve quality of life for those not affected by COVID-19. Future care should be well-balanced between patient groups and preferentially aimed at those at highest risk. Ongoing and future studies can hopefully more robustly define modifiable predictors of poor psychological recovery and its more elaborate risk stratification, to align and improve subsequent targeted implementation of in-hospital and aftercare initiatives.

## Supporting information

**S1 Table. Possible predictors for psychological distress.**
(DOCX)

**S2 Table. Reliability of the impact of event scale-revised, the hospital anxiety and depression scale, the short-form 36, and the EQ-5D in the entire population and stratifications.**
(DOCX)

**S3 Table. Overall response rate and response rates for the individual questionnaires at both follow-up time-points.**
(DOCX)

**S4 Table. Non-responders' analysis based on baseline demographics and treatment-related characteristics.**
(DOCX)

**S5 Table. Comparison of baseline demographics and treatment-related characteristics between full responders and participants who only responded at 1 of 3 months after discharge.**
(DOCX)

**S6 Table. Comparison psychological and HRQoL outcomes between full responders and participants who only responded at 1 of 3 months after discharge.**
(DOCX)

**S7 Table. Non-responders' analysis based on psychological outcomes and health-related quality of life.**
(DOCX)

**S8 Table. Baseline demographics and treatment-related characteristics of study participants.**
(DOCX)

**S9 Table. Analysis of predictors for the development of probable PTSD, anxiety, and depression, and the self-perceived health state.**
(DOCX)

**S1 File. Registered study protocol.**
(DOCX)

**S2 File. Strobe 2017 statement–checklist of items that should be included in reports of cohort studies.**
(DOCX)

**S3 File. Raw data file.**
(XLSX)

## Acknowledgments

We thank the participants in this trial. We also thank the medical interns recruited by the Franciscus Academy for their contributions in the data collection of the Franciscus Corona Registry Database.

## Author Contributions

**Conceptualization:** Johan Hendrik Vlake, Michel Egide van Genderen.

**Formal analysis:** Johan Hendrik Vlake, Sanne Wesselius, Bianca Boxma-de Klerk.

**Investigation:** Johan Hendrik Vlake, Sanne Wesselius.

**Methodology:** Johan Hendrik Vlake, Michel Egide van Genderen, Jasper van Bommel, Bianca Boxma-de Klerk, Evert-Jan Wils.

**Project administration:** Johan Hendrik Vlake, Sanne Wesselius.

**Resources:** Sanne Wesselius.

**Supervision:** Johan Hendrik Vlake, Michel Egide van Genderen, Jasper van Bommel, Evert-Jan Wils.

**Validation:** Johan Hendrik Vlake, Bianca Boxma-de Klerk, Evert-Jan Wils.

**Writing – original draft:** Johan Hendrik Vlake, Sanne Wesselius, Evert-Jan Wils.

**Writing – review & editing:** Johan Hendrik Vlake, Sanne Wesselius, Michel Egide van Genderen, Jasper van Bommel, Bianca Boxma-de Klerk, Evert-Jan Wils.

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
