## [Decision Letter · Decision Letter 0]

27 Jan 2021

PONE-D-20-38679

Psychological distress and health-related quality of life in patients after hospitalization during the COVID-19 pandemic: a single-centre, observational study.

PLOS ONE

Dear Dr. Vlake,

Thank you for submitting your manuscript to PLOS ONE. After careful consideration, we feel that it has merit but does not fully meet PLOS ONE’s publication criteria as it currently stands. Therefore, we invite you to submit a revised version of the manuscript that addresses the points raised during the review process.

I have read your interesting manuscript with pleasure and received the comments of three independent reviewers. The critical, detailed, very clear and helpful comments of the reviewers are included in this letter. I fully agree and share their comments and will try not repeat them in my comments below.

My general impression is that you have so many study variables you all want to use (among a relatively small group), that the condensed format you have chosen to report your findings in, hinders (among other things) the readability of your manuscript. I would like to suggest to rethink your analyses: your main aim seems to be the assessment of the course of PTSD, anxiety and depression symptom, and quality of life among different patient groups over two waves. One major problem you face is the non-response at T1 or T2 which hinders clarity about your study sample (and N). Using mixed effects models could be a way to solve this problem (or using multiple imputation). Of course, you can enter relevant covariates in your analyses. Perhaps covid and non-covid groups could be merged to limit the number of groups/comparisons (with IC as dummy variable). Your second aim seems to be the assessment of predictors of your dependent variables. As also mentioned by the reviewers, this element in your manuscript lacks clarity and focus. It seems more data driven than anything else (see long list of correlations). I would like to suggest to limit the number of predictors, and use current knowledge on predictors or PTSD, etc to choose a limit number of relevant predictors. There are several reviews and meta-analyses available that are very helpful in this perspective.  
Your very short introduction section lacks a proper overview of what is known about the effect is illness/ hospitalization on mental health (including PTSD), and introduction of study variables. This must be added to be able to understand your analyses.
According to your manuscript, 623 patients were eligible and 171 participated in both surveys. The non-response should be included in the text before the results section. In general, the manuscript lacks a clear overview in the text and figures about the number of respondents the analyses and figures refer to. This need to be solved.  You compared subgroups of patients (see also figures) while at the same time you showed that these subgroups differed at baseline. In your analyses you did not control for these differences which is not allowed. Your figures in figure 3 in which you compare groups must therefore be omitted. The added value of your figures was unclear to me.
The outcome measures need to be clarified much more in detail (including adding cronbachs alphas for all assessments at the two waves). You did not examine mental disorders using the IESR and HADS. Please define high scores as probable PTSD etc. and avoid the use of the term disorder in this perspective. You used the Bonferroni correction without explaining why. Without a proper explanation I urge you to drop this often misused correction method. Please consult papers on this issue or Lakens´ blog. He wrote an interesting piece about this correction. http://daniellakens.blogspot.com/2016/02/why-you-dont-need-to-adjust-you-alpha.html Because of all the serious issues raised by the reviewers and my ow comments that need to be addressed first, it is not useful to comment of the discussion section at this stage.

If you would like to make changes to your financial disclosure, please include your updated statement in your cover ltter. Guidelines for resubmitting your figure files are available below the reviewer comments at the end of this letter.

We look forward to receiving your revised manuscript.

Kind regards,

Peter G. van der Velden, Ph.D.

Academic Editor

PLOS ONE

Journal Requirements:

2.We note that you have indicated that data from this study are available upon request. PLOS only allows data to be available upon request if there are legal or ethical restrictions on sharing data publicly. For information on unacceptable data access restrictions, please see http://journals.plos.org/plosone/s/data-availability#loc-unacceptable-data-access-restrictions.

Reviewers' comments:

Reviewer's Responses to Questions

**Comments to the Author**

1. Is the manuscript technically sound, and do the data support the conclusions?

Reviewer #1: Yes

Reviewer #2: Partly

Reviewer #3: Partly

2. Has the statistical analysis been performed appropriately and rigorously? 

Reviewer #1: No

Reviewer #2: No

Reviewer #3: Yes

3. Have the authors made all data underlying the findings in their manuscript fully available?

Reviewer #1: No

Reviewer #2: Yes

Reviewer #3: No

4. Is the manuscript presented in an intelligible fashion and written in standard English?

Reviewer #1: Yes

Reviewer #2: Yes

Reviewer #3: No

5. Review Comments to the Author

Reviewer #1: The article present san interesting piece of research of psychological reactions and symptoms of hospitalized patients with COVID symptoms. The literature review is clear, and the authors present the methods in an adequate way. However, there are two main issue with the paper that I think might be considered by the authors regard to description of what they are studying and the report of results. The authors declare that they are presenting data for psychological disorders, and not what they are presenting is symptoms reported by patients, more information is needed to arrive to diagnosis, so I suggest changing that presentation of the results.

Moreover, I think the methods section can include more information and justification of the decisions made. It is important that the authors declare why they choose one and three months as appropriate time points to collect the data. In the statistical analysis description, authors must include a justification of why to choose variables with correlations with a p-value below 0.10 to be included in a multivariate linear regression model. Moreover, it is important to justify why they used a multivariate linear regression for this purpose, the other analysis suggest that the data do not follow a parametric distribution.

In the results section, the confusion in the section "Predictors of psychological outcomes and quality of life" remains as a result of the lack of the explanation in the statistical analysis section. If the authors maintain the idea to include de multivariate linear regression, I suggest first to declare, in the paper, which variables will be included in the model, and then to present the results of the significance of the model and betas, SE and p values for all the variables.

Also, in the results section, in line 125 the authors indicate that 24 of the 201patients reported a history of mental illness. Please clarify why only there is information about this issue of 201 patients of the 282 patients that participate. If they rest of participants did not respond to that question, please state that in the article.

Reviewer #2: Review:

This study concerns a very interesting topic, which has not yet received much attention in research yet. There are, however, some major points I feel should be addressed to make it ready for publication.

Major points:

1. The introduction is very brief, and fails to adequately emphasize the relevance of the study: what does it add to what is known, why is it important to study this.

2. More importantly, no explanation is given for assumed relationships. Please provide theoretical framework for the described relationships. For instance: why do medical problems that require hospitalization lead to psychological problems? What reasons do you have to believe the relationship is causal? The most striking example of this is the hypothesis formulated on page 3. Absolutely no theoretical (nor empirical) foundation for this is given. Why do you expect this? What is the mechanism behind this?

3. The article generally seems to lack focus. Please explain in your introduction why the (primary and secondary) outcomes are important to study during the COVID pandemic. Do you expect this to defer from regular outcomes (outside pandemics) and for all patients admitted with covid-like symptoms or just for the confirmed covid cases? Why? Elaborate. HRQOL and work-resumption suddenly appear when you describe the aims of the study but not before (work-resumption is also completely ignored in the abstract). Why study these?

4. The study has no pre-exposure measurement. The degree to which participants suffered from some degree of psychological problems before hospitalization As a result whether psychological symptoms are caused by hospitalization during the covid-19 pandemic is unknown. There are even studies that suggest that having pre-existing psychological disorders can be a risk factor for contracting COVID-19 (see https://www.thelancet.com/journals/lanpsy/article/PIIS2215-0366(20)30462-4/fulltext#:~:text=A%20meta%2Danalysis%20of%20pooled,%2C%20poor%20memory%2C%20and%20insomnia).

5. No proper distinction is made between posttraumatic stress SYMPTOMS and posttraumatic stress DISORDER. The IES-r is not able to make a diagnosis. It is therefore common practice to speak of (high levels of) PTSS, or probable PTSD. Furthermore, if you really wanted to examine predictors of PTSD, not PTSS a linear regression is not the correct analysis, after all, one either has PTSD or not.

6. Relate outcome levels to outcome levels outside pandemic conditions in previous studies. Can we conclude that there is a problem specific to pandemics?

7. Educational level is added to the regression models as a linear variable instead of dummy variables. This is a categorical variable with unequal distances between categories. The resulting coefficient is therefore meaningless.

8. The description of the measured used in the study is not sufficient. Please add range of scales and reliability statistics within your sample. Also elaborate on operationalization of other variables such as demographics. For instance, how is ethnicity operationalized? If more than two categories are used in the regression (such as native and non-native) see my previous comment as well.

9. No distinction is made in the regression analyses between the subgroups. Theoretically it would be interesting to distinguish the COVID and non-COVID patients. Ideally this would be done using a multilevel model to determine the impact of group membership on the influence of predictors, but the samples are probable too small for this.

10. I think the statement on p16 that “Collectively, this suggest that participating patients are grossly representative for the total cohort” is too bold. Your response rate is quite low, and examining patient records do not inform you how they would have scored on the outcome measures. A nice idea to ask a small sample of non-responders to respond anyway, but self-selection here is of course also a factor. You could consider comparing outcomes between the full responders (T1 and T2) and those that responded at T1 only.

Minor points:

1. I prefer a table where all primary and secondary outcomes can be seen, instead of a series of boxplots (which seem rather archaic). Having to look up a long list of supplementary tables is not helping your readers understand your work. especially as the tables put in the supplementary section are core study results. Readers are now forced to resort to the long summary of the results in the text, which do not give all information. I propose to put them into the article, and to merge them, and move the box-and-whisker plots to the supplementary section for the fans. This also means shortening the results section as not all results in the table then need to be described in text.

2. The Venn-diagrams look nice, but also add little. Please put these in the supplementary section as well.

3. There is no table for your regression analyses, not even in the supplements! Please add these (to the article, not the supplementary section).

4. Please check for spelling and grammar errors such as “Predictors off psychological outcomes” on page 12. Or also see comment 10 these suggest/this suggests.

5. When describing outcomes (especially when tables are not included) please add direction of association. E.g.: does having had a job before hospitalization lead to higher or lower psych symptoms?

6. Page 3: “the pandemic peak”. Is no longer applicable as there are 2 peaks at this point in time.

7. Provide a benchmark for HRQOL. You state it is low, but what are reference levels for for instance the general population?

Reviewer #3: The study topic “Psychological distress and health-related quality of life in patients after hospitalization during the COVID-19 pandemic: a single-centre, observational study” is relevant and interesting in this era of COVID-19 but is not clearly reported. I thank the authors for this good research, however many things need to be worked on to reach a level of publication

6. PLOS authors have the option to publish the peer review history of their article (what does this mean?). If published, this will include your full peer review and any attached files.

Reviewer #1: No

Reviewer #2: No

Reviewer #3: **Yes: **Desire Aime Nshimirimana

---

## [Author Response · Author response to Decision Letter 0]

24 Mar 2021

Response to the handling editor:

Dear Dr. Vlake,

Thank you for submitting your manuscript to PLOS ONE. After careful consideration, we feel that it has merit but does not fully meet PLOS ONE’s publication criteria as it currently stands. Therefore, we invite you to submit a revised version of the manuscript that addresses the points raised during the review process.

I have read your interesting manuscript with pleasure and received the comments of three independent reviewers. The critical, detailed, very clear and helpful comments of the reviewers are included in this letter. I fully agree and share their comments and will try not repeat them in my comments below.

RESPONSE: 

Dear Dr. Peter G. van der Velden,

We really appreciate that you are handling our manuscript, and that you are willing to give us the opportunity to thoroughly revise our manuscript. The suggestions, concerns, and comments raised gave us the opportunity to further improve our manuscript, and we have made point-by-point responses to all comments below.

For the referral to certain pages or lines within the manuscript; all referrals are based on the submitted marked-up manuscript, in which the changes made during the revision are highlighted.

1) My general impression is that you have so many study variables you all want to use (among a relatively small group), that the condensed format you have chosen to report your findings in, hinders (among other things) the readability of your manuscript. I would like to suggest to rethink your analyses: your main aim seems to be the assessment of the course of PTSD, anxiety and depression symptom, and quality of life among different patient groups over two waves. One major problem you face is the non-response at T1 or T2 which hinders clarity about your study sample (and N). Using mixed effects models could be a way to solve this problem (or using multiple imputation). Of course, you can enter relevant covariates in your analyses. Perhaps covid and non-covid groups could be merged to limit the number of groups/comparisons (with IC as dummy variable). Your second aim seems to be the assessment of predictors of your dependent variables. As also mentioned by the reviewers, this element in your manuscript lacks clarity and focus. It seems more data driven than anything else (see long list of correlations). I would like to suggest to limit the number of predictors, and use current knowledge on predictors or PTSD, etc to choose a limit number of relevant predictors. There are several reviews and meta-analyses available that are very helpful in this perspective. 

RESPONSE: 

We understand the comment raised, and in hindsight agree that we shared a too large amount of information within the paper, and that this hindered the readability of the manuscript. The main aim of the study was indeed to assess the course of psychological sequelae (symptoms of PTSD, anxiety, and depression) and quality of life in time after hospital discharge in patients included during the first COVID-19 pandemic wave. To address this comment, we have made the following changes:

1) We adjusted figure 1 to make it more insightful how inclusion was performed.

2) As suggested by the editor, we have re-analyzed the entire results section using regression models, adjusting for baseline differences. In order to determine possible predictors, we have used linear and logistic mixed models.

We therefore added:

“Differences in outcome measures between stratifications were analyzed using linear or logistic regression models for continuous and categorical outcomes, respectively, adjusting for baseline differences between groups, in which the ‘stratification’ and at baseline differing characteristics served as independent variables.” (Methods, page 9, lines 172-174).

“To identify possible predictors for psychological symptomatology or quality of life, we first conducted a univariate linear or logistic mixed model with possible predictors as independent variables (S1. Table). Variables with a p-value ≤0.10 were added in the multivariate linear or logistic mixed model to determine predictors.” (Methods, page 9, lines 180-182)

“Psychological outcomes of randomly selected non-responders were compared with those of responders using a linear or logistic regression model, adjusting for baseline differences between groups, in which the ‘stratification’ and at baseline differing characteristics served as independent variables.” (Methods, page 9, lines 184-186)

Additionally, we changed the Results section within the Abstract section accordingly:

“282 of 623 eligible patients participated in this study (median age 64 years, 36% female). 16% and 12% of these patients reported clinically relevant symptoms of PTSD, 29% and 20% clinically relevant symptoms of anxiety, and 32% and 25% clinically relevant symptoms of depression after one and three months, respectively. ICU patients reported less probable depression, but no differences were found in PTSD, anxiety, or overall HRQoL. COVID-19 patients had a worse physical and mental quality of life one month after discharge, and ICU patients reported a better physical quality of life three months after discharge. PTSD severity was predicted by time after discharge and not being Caucasian. Severity of anxiety was predicted by time after discharge, age, educational level, employment status, COVID-19 diagnosis, and not being Caucasian. Depression severity was predicted by time after discharge and educational level. 49% and 76% of previously working patients had resumed work, and patients worked a median of 30 hours less at both follow-up time-points. COVID-19 patients more frequently experienced a financial decline. ” (Abstract, page 2-3, lines 40-49)

3) We limited the number of variables included for the prediction model, and these variables were selected based on existing literature on risk factors for developing psychological sequelae after hospitalization. We therefore added the following sentences into the methods section:

“These predictors were chosen based on previous literature, and included age, gender, ethnicity, educational level, duration of admission, ICU admission, COVID-19 diagnosis and work before admission, and severity of disease in terms of SOFA score at the first day of COVID-19 suspicion (day of enrolment). Literature supporting these predictors is depicted in S1. Table in the Supporting Information.” (Methods, page 8, lines 158-162)

2) Your very short introduction section lacks a proper overview of what is known about the effect is illness/ hospitalization on mental health (including PTSD), and introduction of study variables. This must be added to be able to understand your analyses.

RESPONSE: 

We agree with the editor that the introduction sections in its present state is limited, and therefore added information about the effect of illness/hospitalization on mental health and the rationale for using certain variables as predictors for psychological distress and quality of life.

Concordantly, we added the following:

“Symptoms of the infectious disease caused by the SARS-CoV-2 virus, coronavirus disease-19 (COVID-19), vary from mild respiratory and gastro-intestinal symptoms to severe acute respiratory failure, requiring respiratory support in an Intensive Care Unit (ICU) (2-4).” (Introduction, page 4, lines 64-66)

“…; symptoms of depression and anxiety occur in 12-67% of hospitalized patients; symptoms of post-traumatic stress disorder (PTSD) in 19%-45% (8-15). Survivors of ICU treatment are known to be at risk for psychological impairments including symptoms of depression, anxiety and PTSD. These symptoms occur in up to 20%-67% of ICU survivors (16-21). Post-hospitalization psychological impairments negatively impact on the ability to return to work and the health-related quality of life (HRQoL) (13, 17, 18, 20).” (Introduction, page 4, lines 77-81)

“The risk of psychological impairments is expected to be even higher in patients hospitalized during the COIVD-19 pandemic due to the exceptional circumstances and uncertainties (22, 23).“ (Introduction, page 4, lines 82-83)

3) According to your manuscript, 623 patients were eligible and 171 participated in both surveys. The non-response should be included in the text before the results section. In general, the manuscript lacks a clear overview in the text and figures about the number of respondents the analyses and figures refer to. This need to be solved. 

RESPONSE: 

We agree that the manuscript in its present state lacks this clear overview. We therefore rewrote the results section, expanded the introduction, and changes the figures and tables in the text, and removed several tables from the supplementary materials to overcome this issue.

With regard to the inclusion rate; we adjusted figure 1 to make it more insightful. We also replaced the paragraph on this matter prior to the ‘Participants’ section within the Results section. In addition, we addressed the non-inclusion rate by adding the following sentence to the result section:

“Non-participating patient either declined participation (n=262) or did not respond to the questionnaires or to the reminding phone calls (n=79).” (Results, page 10, lines 195-197)

4) You compared subgroups of patients (see also figures) while at the same time you showed that these subgroups differed at baseline. In your analyses you did not control for these differences which is not allowed. Your figures in figure 3 in which you compare groups must therefore be omitted. The added value of your figures was unclear to me.

RESPONSE:

We agree with the editor’s comment that we should control for baseline differences. In line with the response to the editor’s previous comments, we changed the analysis within the results section, and used regression and mixed models instead of non-controlling tests (i.e., Wilcoxon or t-test or T-test).

With regard to the Figure 3 and Figure 4, in line with the comment raised by the editor, we have omitted these figures, and prioritized tables instead.

5) The outcome measures need to be clarified much more in detail (including adding cronbachs alphas for all assessments at the two waves). You did not examine mental disorders using the IESR and HADS. Please define high scores as probable PTSD etc. and avoid the use of the term disorder in this perspective.

RESPONSE: 

We agree with the editor that the description of the used questionnaires was limited, and, in accordance with the raised concern, we described the outcome measures in more detail.

Therefore we added the following sentences:

“The IES-R is a validated 22-item questionnaire that quantifies the subjective distress a person is experience after a traumatic event (32). The IES-R yields a total score, ranging from 0-88 (higher scores indicating more severe symptoms), and subscale scores can be calculated for symptoms of intrusion, avoidance and hyperarousal.” (Methods, page 7, lines 132-135)

“The HADS is commonly used to determine the levels of anxiety and depression that a person is experiencing (34). The HADS is 14-item scale that generated ordinal data. Seven of the items relate to anxiety and seven relate to depression. The HADS yields a depression and anxiety sum score, ranging from 0 to 21 with higher scores indicating more severe symptoms.” (Methods, page 7, lines 135-138)

“The EQ-5D measures the HRQoL on five dimensions (mobility, self-care, usual activities, pain/discomfort and anxiety/depression), by which the weight of a health state can be computed into an EQ-5D Time Trade Off (TTO) score. This score ranges from – 0.446 (worst quality of life) to 1.000 (best quality of life). Additionally, patients score their current subjective health state on a visual analogue scale (the EQ-VAS), ranging from 0 (worst health imaginable) to 100 (best health imaginable) (36, 37).” (Methods, page 7-8, lines 141-145)

“The RAND-36 is a 36-item, self-reported survey of HRQoL, and consists nine scales scores, which are the weighted sums of the questions in their section (38). Each scale is directly transformed to a scale ranging from 0 (worse score) to 100 (best score) on the assumption that each question carries an equal weight. The nine scores are; …” (Methods, page 8, lines 148-150)

“Resumption to work and financial decline due to the hospitalization were was additionally assessed using self-composed questionnaires.” (Methods, page 8, lines 154-155)

The editor also rightly point out that we did not determine the presence of psychiatric disorders, but rather the presence of symptomatology, and thus the probability of the disorders. We therefore removed ‘disorders’ from the manuscript, and used probable PTSD, anxiety, and depression instead.

6) You used the Bonferroni correction without explaining why. Without a proper explanation I urge you to drop this often misused correction method. Please consult papers on this issue or Lakens´ blog. He wrote an interesting piece about this correction. http://daniellakens.blogspot.com/2016/02/why-you-dont-need-to-adjust-you-alpha.html

RESPONSE: 

We agree with the editor that the use of the Bonferroni correction was lacking. As we changed our analysis by using regression and mixed models, we did no longer need correction and therefore deleted the following from the manuscript:

“We corrected for multiple testing by using a Bonferroni correction.”

7) Because of all the serious issues raised by the reviewers and my ow comments that need to be addressed first, it is not useful to comment of the discussion section at this stage.

RESPONSE: 

We understand the editor that reviewing the discussion at this stage was not useful. Of note, we revised our discussion based on the revisions, and hope that our revisions are sufficient to review our discussion section.

8) Please submit your revised manuscript by Mar 13 2021 11:59PM. If you will need more time than this to complete your revisions, please reply to this message or contact the journal office at plosone@plos.org. RESPONSE: 

We have submitted the revised manuscript prior to the assigned deadline.

RESPONSE: 

The current document responds to the points raised by the academic editor and reviewers.

RESPONSE: 

We submitted a marked-up copy of the manuscript highlighting all chances made to the manuscript.

RESPONSE: 

We have submitted a clean version of the manuscript, without tracked changes, to the revised submission of the manuscript

Response to the Reviewer's Responses to Questions

1. Is the manuscript technically sound, and do the data support the conclusions?

Reviewer #1: Yes

Reviewer #2: Partly

Reviewer #3: Partly

RESPONSE: 

We have changed the introduction, methods, and results according to the comments raised by the reviewers, and hope this has resulted in the manuscript being technically sound, and to more clearly illustrate the data supporting the conclusions.

2. Has the statistical analysis been performed appropriately and rigorously? 

Reviewer #1: No

Reviewer #2: No

Reviewer #3: Yes

RESPONSE:

We have changed the analysis methods by using regression and mixed models. We thereby hope that the statistical analysis has been performed appropriately and rigorously.

3. Have the authors made all data underlying the findings in their manuscript fully available?

Reviewer #1: No

Reviewer #2: Yes

Reviewer #3: No

RESPONSE: 

We have submitted the de-identified data as supporting information, and thereby made all data underlying the findings in the manuscript fully available.

We added the following to the Data Sharing Statement in the Declarations:

“The datasets used and/or analyzed during the current study are included in the supporting information files.“ (Page 27, line 485)

4. Is the manuscript presented in an intelligible fashion and written in standard English?

Reviewer #1: Yes

Reviewer #2: Yes

Reviewer #3: No

RESPONSE: 

We have copy-edited the entire manuscript to ensure it is presented in an intelligible fashion and written in Standard English.

Response to Comments raised by Reviewer #1:

The article present san interesting piece of research of psychological reactions and symptoms of hospitalized patients with COVID symptoms. The literature review is clear, and the authors present the methods in an adequate way. 

1. However, there are two main issue with the paper that I think might be considered by the authors regard to description of what they are studying and the report of results. The authors declare that they are presenting data for psychological disorders, and not what they are presenting is symptoms reported by patients, more information is needed to arrive to diagnosis, so I suggest changing that presentation of the results.

RESPONSE: 

We thank you for reviewing our manuscript, and for your useful comments. We agree that we should not have used the term ‘disorder’, a similar concern was raised by the academic editor (comment 5). We therefore discarded ‘disorder’ from the manuscript and changed it with probable PTSD, anxiety, or depression.

2. Moreover, I think the methods section can include more information and justification of the decisions made. It is important that the authors declare why they choose one and three months as appropriate time points to collect the data. 

RESPONSE: 

We agree with the reviewer that the methods section should have included more information. A similar comment was raised by the academic editor (comment 5).

With regard to the chosen time-points to collect the data, we added the following to the manuscript:

“Psychological outcomes were assessed one month and three months after hospital discharge to 1) compare results with other research concerning psychological outcomes after hospital admission due to COVID-19 or similar coronaviruses; studies up till now mainly reported data of the first month after discharge (27) and 2) to evaluate the course of psychological symptoms and HRQoL in time after discharge.“ (Methods, page 8, lines 163-167)

3. In the statistical analysis description, authors must include a justification of why to choose variables with correlations with a p-value below 0.10 to be included in a multivariate linear regression model. 

RESPONSE: 

First, based on the comment raised by the academic editor (comment 1), we have changed our analysis using mixed models for the identification of possible predictors. Therefore, we have analyzed all possible predictors using a univariate mixed model and used the variables with a p-value ≤0.10 for the full (multivariate) mixed model. Thereby, we have discarded the use of correlations from original manuscript.

Within prediction models, there are several manners to select variables, of which one it to perform univariate analyses on each variables and include those with a small (≤0.10) p-value into multivariate (mixed) models. 

Concordantly, we added the following:

“Differences in outcome measures between stratifications were analyzed using linear or logistic regression models for continuous and categorical outcomes, respectively, adjusting for baseline differences between groups, in which the ‘stratification’ and at baseline differing characteristics served as independent variables.” (Methods, page 9, lines 172-174)

“To identify possible predictors for psychological symptomatology or quality of life, we first conducted a univariate linear or logistic mixed model with possible predictors as independent variables (S1. Table). Variables with a p-value ≤0.10 were added in the multivariate linear or logistic mixed model to determine predictors.” (Methods, page 9, lines 180-182)

4. Moreover, it is important to justify why they used a multivariate linear regression for this purpose, the other analysis suggest that the data do not follow a parametric distribution.

RESPONSE: 

We agree with the reviewer that using a multivariate linear regression model was a suboptimal choice. We have changed the analysis within the manuscript, and now used mixed models instead, of which the assumptions rely less on the distribution of the variables within the model. For the changes we made to the statistical analysis, see our response to your previous comment.

5. In the results section, the confusion in the section "Predictors of psychological outcomes and quality of life" remains as a result of the lack of the explanation in the statistical analysis section. If the authors maintain the idea to include de multivariate linear regression, I suggest first to declare, in the paper, which variables will be included in the model, and then to present the results of the significance of the model and betas, SE and p values for all the variables.

RESPONSE: 

We agree with the reviewer that we did not explain our choice of variable, nor the analysis of predictors, well enough. A similar comment was raised by the academic editor (comment 1).

With regard to the choice of predictive variables, we have added the following sentences into the manuscript:

“These predictors were chosen based on previous literature, and included age, gender, ethnicity, educational level, duration of admission, ICU admission, COVID-19 diagnosis and work before admission, and severity of disease in terms of SOFA score at the first day of COVID-19 suspicion (day of enrolment). Literature supporting these predictors is depicted in S1. Table in the Supporting Information.” (Methods, page 8, lines 158-162).

In addition, in the results section, we have added the betas, SE, and p-values of all significant variables used in the models, and added tables depicting the outcomes of the univariate and multivariate mixed models, including betas and p-values, to predict the severity of psychological symptoms and HRQoL in table 4 and 5 in the manuscript, and the tables concerning the predictors of the prevalence of psychological symptoms in table S3.

6. Also, in the results section, in line 125 the authors indicate that 24 of the 201patients reported a history of mental illness. Please clarify why only there is information about this issue of 201 patients of the 282 patients that participate. If they rest of participants did not respond to that question, please state that in the article.

RESPONSE: 

We agree with the reviewer that this is not well described in the manuscript. Patients were indeed asked about their mental history and could self-determine whether they wanted to share this history with us. 

Therefore, we added the following sentence into the manuscript:

“Patients were asked for their psychiatric history and could choose whether or not to answer.” (Methods, page 8, line 155-156).

“…, who were willing to share this information.” (Results, page 10, line 204).

Response to Comments raised by Reviewer #2:

This study concerns a very interesting topic, which has not yet received much attention in research yet. There are, however, some major points I feel should be addressed to make it ready for publication.

Major points:

1. The introduction is very brief, and fails to adequately emphasize the relevance of the study: what does it add to what is known, why is it important to study this.

RESPONSE: 

We agree with the reviewer that the introduction was to brief, and a similar concern was raised by the academic editor (comment 2). We therefore expanded the introduction, as further described in our response to comment 2 of the academic editor.

2. More importantly, no explanation is given for assumed relationships. Please provide theoretical framework for the described relationships. For instance: why do medical problems that require hospitalization lead to psychological problems? What reasons do you have to believe the relationship is causal? The most striking example of this is the hypothesis formulated on page 3. Absolutely no theoretical (nor empirical) foundation for this is given. Why do you expect this? What is the mechanism behind this?

RESPONSE: 

We agree with the reviewer that more information should be added to the manuscript on the rationale and choices made within our study.

To the introduction, we added the following:

“…; symptoms of depression and anxiety occur in 12-67% of hospitalized patients; symptoms of post-traumatic stress disorder (PTSD) in 19%-45% (8-15). Survivors of ICU treatment are known to be at risk for psychological impairments including symptoms of depression, anxiety and PTSD. These symptoms occur in up to 20%-67% of ICU survivors (16-21). Post-hospitalization psychological impairments negatively impact on the ability to return to work and the health-related quality of life (HRQoL) (13, 17, 18, 20).” (Introduction, page 4, lines 77-81)

“The risk of psychological impairments is expected to be even higher in patients hospitalized during the COIVD-19 pandemic due to the exceptional circumstances and uncertainties (22, 23).“ (Introduction, page 4, lines 82-83)

Additionally, we added the following to the methods:

“These predictors were chosen based on previous literature, and included age, gender, ethnicity, educational level, duration of admission, ICU admission, COVID-19 diagnosis and work before admission, and severity of disease in terms of SOFA score at the first day of COVID-19 suspicion (day of enrolment). Literature supporting these predictors is depicted in S1. Table in the Supporting Information.” (Methods, page 8, lines 158-162).

3. The article generally seems to lack focus. Please explain in your introduction why the (primary and secondary) outcomes are important to study during the COVID pandemic. Do you expect this to defer from regular outcomes (outside pandemics) and for all patients admitted with covid-like symptoms or just for the confirmed covid cases? Why? Elaborate. HRQOL and work-resumption suddenly appear when you describe the aims of the study but not before (work-resumption is also completely ignored in the abstract). Why study these?

RESPONSE: 

We thank the reviewer for this comment. First, we added the following to the introduction to emphasize that we hypothesize that patients, as in our cohort, have an increased risk of developing psychological symptomatology:

“…; symptoms of depression and anxiety occur in 12-67% of hospitalized patients; symptoms of post-traumatic stress disorder (PTSD) in 19%-45% (8-15). Survivors of ICU treatment are known to be at risk for psychological impairments including symptoms of depression, anxiety and PTSD. These symptoms occur in up to 20%-67% of ICU survivors (16-21). ” (Introduction, page 4, lines 77-81)

“The risk of psychological impairments is expected to be even higher in patients hospitalized during the COIVD-19 pandemic due to the exceptional circumstances and uncertainties (22, 23).“ (Introduction, page 4, lines 81-82)

In addition, we have added the following to explain why we also study HRQoL and work resumption:

“Post-hospitalization psychological impairments negatively impact on the ability to return to work and the health-related quality of life (HRQoL) (13, 17, 18, 20).” (Introduction, page 4, lines 80-81)

We added the following to the abstract, as this point was indeed missing in the abstract of our original manuscript:

“Illnesses requiring hospitalization are known to negatively impact psychological well-being and health-related quality of life (HRQoL) after discharge.” (Abstract, page 2, lines 24-25)

“…. is expected to be higher due to the exceptional circumstances within and outside the hospital during the pandemic surge ” (Abstract, page 2, lines 26-27)

“We also aimed to determine HRQoL and work resumption, to explore predictors for psychological distress and HRQoL, and to examine whether psychological distress was higher in COVID-19 confirmed patients, and in those treated in ICUs.“ (Abstract, page 2, lines 29-31)

“Health-related quality of life (HRQoL) and work resumption were secondary outcomes. Predictors for psychological distress and HRQoL were explorative outcomes.” (Abstract, page 2, lines 36-37)

“PTSD severity was predicted by time after discharge and not being Caucasian. Severity of anxiety was predicted by time after discharge, age, educational level, employment status, COVID-19 diagnosis, and not being Caucasian. Depression severity was predicted by time after discharge and educational level. 49% and 76% of previously working patients had resumed work, and patients worked a median of 30 hours less at both follow-up time-points. COVID-19 patients more frequently experienced a financial decline.“ (Abstract, page 2, lines 45-49)

4. The study has no pre-exposure measurement. The degree to which participants suffered from some degree of psychological problems before hospitalization As a result whether psychological symptoms are caused by hospitalization during the covid-19 pandemic is unknown. There are even studies that suggest that having pre-existing psychological disorders can be a risk factor for contracting COVID-19 (see https://www.thelancet.com/journals/lanpsy/article/PIIS2215-0366(20)30462-4/fulltext#:~:text=A%20meta%2Danalysis%20of%20pooled,%2C%20poor%20memory%2C%20and%20insomnia).

RESPONSE: 

We agree with the reviewer that the lack of a pre-exposure measure is missing. A pre-exposure measure was unfortunately not attainable giving the study circumstances. However, as recently discussed by Geense et al., pre-exposure in unplanned admitted patients (all of the included patients in our cohort), depends on retrospective recollection of psychological functioning and HRQoL. Moreover, pre-exposure PTSD scoring is meaningless, as PTSD is inherently related to the traumatizing event under study. Notwithstanding its shortcomings, we strongly agree with the reviewer that future studies on psychological and HRQoL recovery should strive for pre-exposure measure. 

We addressed this comment in the discussion section:

“First, non-COVID-19 patients suffered from more comorbidities and were older, indicative of a worse pre-existent health status, but baseline psychological status and HRQoL of our cohort were unfortunately not available. We thus cannot formally rule out that the pre-existing psychological well-being was poorer and resilience lower, predisposing for impaired psychological recovery (47).” (Discussion, page 24, lines 410-413)

Additionally, we added the following to the discussion section to emphasize this limitation;

“First, the high incidence of psychological symptomatology and the poor HRQoL post-discharge may be either attributed to factors related to hospitalization or to baseline psychological imbalances. Baseline psychological status and HRQoL in our cohort was not available, but only a few patients in our cohort reported a history of mental illness, suggesting that hospitalization rather than pre-existing psychological distress relates to post-discharge symptomatology. To overcome this issue, we believe that future studies on psychological and HRQoL recovery should strive for a best effort to obtain a pre-exposure evaluation (53).” (Discussion, page 25, lines 439-445)

5. No proper distinction is made between posttraumatic stress SYMPTOMS and posttraumatic stress DISORDER. The IES-r is not able to make a diagnosis. It is therefore common practice to speak of (high levels of) PTSS, or probable PTSD. Furthermore, if you really wanted to examine predictors of PTSD, not PTSS a linear regression is not the correct analysis, after all, one either has PTSD or not.

RESPONSE: 

We fully agree with the reviewer, and similar concerns were raised by the academic editor and reviewer 2. We have therefore discarded ‘disorder’ from the manuscript, and used probable PTSD, anxiety, and depression.

6. Relate outcome levels to outcome levels outside pandemic conditions in previous studies. Can we conclude that there is a problem specific to pandemics?

RESPONSE: 

We thank the reviewer for this comment but feel that we have addressed these points in the discussion section. 

To summarize these points: In the second paragraph (line 384-393), we compare our results with other studies concerning psychological outcomes during the COVID-19 pandemic.; In the third paragraph (line 394-420), we compare our results with the results from other studies investigating the psychological response to hospitalization in several settings, including those during the MERS and SARS outbreaks.

7. Educational level is added to the regression models as a linear variable instead of dummy variables. This is a categorical variable with unequal distances between categories. The resulting coefficient is therefore meaningless.

RESPONSE: 

We agree with the reviewer and have revised the following to improve this:

1) We added information about the rationale for the possible predictors in the methods section.

2) We used mixed models instead of linear regression models to identify predictors for psychological distress and HRQoL.

3) We added the results from the univariate analysis of all possible predictors, and the full model, in the supplementary information, including betas and p-values.

With regard to the education level variable: we used a dummy instead, and a beta of each category within the variable is presented in the tables.

8. The description of the measured used in the study is not sufficient. Please add range of scales and reliability statistics within your sample. Also elaborate on operationalization of other variables such as demographics. For instance, how is ethnicity operationalized? If more than two categories are used in the regression (such as native and non-native) see my previous comment as well.

RESPONSE: 

We agree with the reviewer that the description of the questionnaires used in this study was limited. We therefore expanded the description of the questionnaires, including the ranges. The reliability of the validated questionnaires used, have been extensively reported previously and we refer to these papers in the method section.

With regard to the demographics, we extracted this information from the electronic patient records and used variable definitions as used for the ISARIC eCRF, as outlined in method section. For table 1, we have used the ethnicity as documented in the electronic patient record, and categorized ethnicity into ‘Caucasian’, ‘Negroid’, ‘Surinamese/Hindustan’, ‘Arab, not specified’, ‘Turkish’, ‘Moroccan’, ‘Other’, and ‘Unknown’. For analyses, we categorized these into ‘Caucasian’ and ‘non-Caucasian’.

9. No distinction is made in the regression analyses between the subgroups. Theoretically it would be interesting to distinguish the COVID and non-COVID patients. Ideally this would be done using a multilevel model to determine the impact of group membership on the influence of predictors, but the samples are probable too small for this.

RESPONSE: 

We have revised our analysis and used mixed models. We have added ‘COVID-19’ and ‘ICU Admission’ as predictors, and only added these to the full model if the univariate analysis had a p-value below 0.10. As such, the predictive value, if present, can be extracted from the multivariate mixed models presented in the supporting information.

10. I think the statement on p16 that “Collectively, this suggest that participating patients are grossly representative for the total cohort” is too bold. Your response rate is quite low and examining patient records do not inform you how they would have scored on the outcome measures. A nice idea to ask a small sample of non-responders to respond anyway, but self-selection here is of course also a factor. You could consider comparing outcomes between the full responders (T1 and T2) and those that responded at T1 only.

RESPONSE: 

We value the suggested outcomes comparison between full responders and those responded at T1 or T2 only as added analysis to address the non-responders’ issue. The results of that comparison is added as Table S8. In summary, the outcomes of full responders versus responders of one time-point only are similar, both for the one- and three-months’ time-point. Moreover, baseline demographic and treatment-related characteristics of these groups were also largely comparable (see Table S7).

We added the following:

“In addition, both baseline demographics and treatment-related characteristics were similar for full responders and those who responded at one month or three months only (S7 Table). No major differences in psychological outcomes and HRQoL were observed between full responders and single time-point responders (S8 table).” (Results, page 22, lines 373-376).

Notwithstanding, we agree with the reviewer that the sentence remains too bold, and we rewrote the sentence;

“…, and in a sample of non-responders’ psychological distress was roughly similar to responders. Collectively, our cohort appears grossly comparable with the overall population at baseline and for psychological outcomes. Baseline characteristics (demographic and treatment-related factors) were similar for non-responders, responders, and a sample of non-responders willing to respond once. In addition, psychological outcomes and HRQoL were comparable for responders and the sample of non-responders willing to respond, and for responders at both time-points as compared to responders at 1 time-point only.” (Discussion, page 25, lines 454-459).

Minor points:

1. I prefer a table where all primary and secondary outcomes can be seen, instead of a series of boxplots (which seem rather archaic). Having to look up a long list of supplementary tables is not helping your readers understand your work. especially as the tables put in the supplementary section are core study results. Readers are now forced to resort to the long summary of the results in the text, which do not give all information. I propose to put them into the article, and to merge them, and move the box-and-whisker plots to the supplementary section for the fans. This also means shortening the results section as not all results in the table then need to be described in text.

RESPONSE: 

We fully agree with the reviewer’s point and have discarded the boxplots, and instead used tables to present our primary and secondary outcomes. We have focused the results section by shortening it, only described relevant outcomes, and referred to the tables for less relevant outcomes.

2. The Venn-diagrams look nice, but also add little. Please put these in the supplementary section as well.

RESPONSE: 

We agree and discarded the Venn-diagrams, because in our new analysis and results section, we do not look into the co-presence of probable PTSD, anxiety, and depression.

3. There is no table for your regression analyses, not even in the supplements! Please add these (to the article, not the supplementary section).

RESPONSE: 

We have added the tables of the mixed model analysis to identify predictors for both the severity of psychological distress (Table 4) and HRQoL (Table 5) to the manuscript, and added the tables with the results of the mixed models for the prediction of the prevalence of probable PTSD, anxiety, and depression to the supplements (S3. Table).

4. Please check for spelling and grammar errors such as “Predictors off psychological outcomes” on page 12. Or also see comment 10 these suggest/this suggests.

RESPONSE: 

We apologize for this untidiness and copy-edited the entire manuscript.

5. When describing outcomes (especially when tables are not included) please add direction of association. E.g.: does having had a job before hospitalization lead to higher or lower psych symptoms?

RESPONSE:

We apologize for this untidiness and agree that this limits the comprehensibility or our results. We therefore added for each predictor whether it increased or reduced the outcome and added the direction by adding the betas.

6. Page 3: “the pandemic peak”. Is no longer applicable as there are 2 peaks at this point in time.

RESPONSE: 

We agree and added ‘first’ to the sentence.

7. Provide a benchmark for HRQOL. You state it is low, but what are reference levels for for instance the general population?

RESPONSE: 

We agree that this information was missing, and added the following to the results:

“Based on the distribution of age in our cohort, the mean TTO score of the Dutch general population is 0.852 (37). A TTO score below 0.852 is considered poor, and a TTO score above 0.852 good.” (Methods, page 8, lines 145-147)

“Based on these scores, a mental and physical component score (MSC, PCS) can be computed for which the mean in the general population will be 50. A MCS or PCS score below 50 is considered low, and a MCS or PCS score above 50 is considered good (39).” (Methods, page 8, lines 152-154)

Response to Comments raised by Reviewer #3:

Reviewer #3: The study topic “Psychological distress and health-related quality of life in patients after hospitalization during the COVID-19 pandemic: a single-centre, observational study” is relevant and interesting in this era of COVID-19 but is not clearly reported. I thank the authors for this good research, however many things need to be worked on to reach a level of publication

Comments to the authors;

“Psychological distress and health-related quality of life in patients after hospitalization during the COVID-19 pandemic: a single-centre, observational study”

The study topic is relevant and interesting but is not clearly reported. 

Introduction 

1. The introduction is too shallow. On top of what is written, the methods of psychological distress measurements used in this study need to be briefly described (i.e PTSD, depression, and anxiety measures using the Impact of Event Scale – Revised (IES-R) and the Hospital Anxiety and Depression Scale (HADS)). Briefly discuss as well how to measure health related quality of life using EQ-5D and RAND-36. This brief description should be done in a very concise and understandable manner. 

RESPONSE: 

Similar comments were raised by the academic editor and the other reviewers.

In short, we have expanded the introduction, and added more background information. We have described the questionnaire used in this study, including the cut-off values and ranges, more thoroughly in the methods section.

In the Introduction, we added the following:

“Symptoms of the infectious disease caused by the SARS-CoV-2 virus, coronavirus disease-19 (COVID-19), vary from mild respiratory and gastro-intestinal symptoms to severe acute respiratory failure, requiring respiratory support in an Intensive Care Unit (ICU) (2-4).“ (Introduction, page 4 lines 64-66)

“; symptoms of depression and anxiety occur in 12-67% of hospitalized patients; symptoms of post-traumatic stress disorder (PTSD) in 19%-45% (8-15). Survivors of ICU treatment are known to be at risk for psychological impairments including symptoms of depression, anxiety and PTSD. These symptoms occur in up to 20%-67% of ICU survivors (16-21). Post-hospitalization psychological impairments negatively impact on the ability to return to work and the health-related quality of life (HRQoL) (13, 17, 18, 20). ” (Introduction, page 4, lines 77-81)

“The risk of psychological impairments is expected to be even higher in patients hospitalized during the COIVD-19 pandemic due to the exceptional circumstances and uncertainties (22, 23).“ (Introduction, page 4, lines 82-83)

In the Methods, we added the following:

“The IES-R is a validated 22-item questionnaire that quantifies the subjective distress a person is experience after a traumatic event (32). The IES-R yields a total score, ranging from 0-88 (higher scores indicating more severe symptoms), and subscale scores can be calculated for symptoms of intrusion, avoidance and hyperarousal.” (Methods, page 7, lines 132-135)

“The HADS is commonly used to determine the levels of anxiety and depression that a person is experiencing (34). The HADS is 14-item scale that generated ordinal data. Seven of the items relate to anxiety and seven relate to depression. The HADS yields a depression and anxiety sum score, ranging from 0 to 21 with higher scores indicating more severe symptoms.” (Methods, page 7, lines 135-138)

“The EQ-5D measures the HRQoL on five dimensions (mobility, self-care, usual activities, pain/discomfort and anxiety/depression), by which the weight of a health state can be computed into an EQ-5D Time Trade Off (TTO) score. This score ranges from – 0.446 (worst quality of life) to 1.000 (best quality of life). Additionally, patients score their current subjective health state on a visual analogue scale (the EQ-VAS), ranging from 0 (worst health imaginable) to 100 (best health imaginable) (36, 37). Based on the distribution of age in our cohort, the mean TTO score of the Dutch general population is 0.852 (37). A TTO score below 0.852 is considered poor, and a TTO score above 0.852 good.” (Methods, page 7, lines 141-147)

“The RAND-36 is a 36-item, self-reported survey of HRQoL, and consists nine scales scores, which are the weighted sums of the questions in their section (38). Each scale is directly transformed to a scale ranging from 0 (worse score) to 100 (best score) on the assumption that each question carries an equal weight. The nine scores are; physical functioning, social functioning, physical role limitations, emotional role limitations, mental health, vitality, pain, general health, and health change. Based on these scores, a mental and physical component score (MSC, PCS) can be computed for which the mean in the general population will be 50. A MCS or PCS score below 50 is considered low, and a MCS or PCS score above 50 is considered good (39). Resumption to work and financial decline due to the hospitalization were was additionally assessed using self-composed questionnaires. Patients were asked for their psychiatric history and could choose whether or not to answer. Baseline characteristics were extracted from electronic patients’ records.” (Methods, page 8, lines 148-157)

2. The problem statement of this study is not clearly stated: i.e “ Data from the previous SARS and MERS epidemics support this concern and more recent studies suggest that up to 50% of COVID-19 patients suffers from psychological distress up to one month after hospital discharge”. Looking at the results of this study (i.e of one of the results; “patients admitted for other illnesses more suffered psychologically than COVID-19” patients) , I was expecting to read a statement contradicting this to support your study results. 

RESPONSE: 

We agree that more elaboration is needed, and added the following sentence in the introduction section:

“However, it is unknown whether this psychological response is attributed to COVID-19 or hospitalization during a pandemic, and whether the psychological response differs between specific subsets of patients, such as non-COVID-19 and ICU patients.” (Introduction, page 4-5, lines 85-87)

We thereby question whether the high incidence of psychological distress during the previous SARS and MERS epidemics, and the incidence of psychological distress in COVID-19 patients is attributed to COVID-19, or to the pandemic circumstances.

3. The title, the aim of the study and results do not link well. Authors also say “Additionally, we tested the hypothesis that the psychological distress was higher in COVID-19 confirmed patients, and in those treated in ICUs”. Where is this hypothesis? Why the authors talk about an hypothesis which is not stated in the study?

RESPONSE: 

We agree that a rationale for this hypothesis was lacking. With the revised introduction, we hope that it has become clearer why we conducted the study. We also described the aim of the study more clearly:

"In this study, we therefore aimed to quantify the psychological distress, i.e., symptoms of PTSD, anxiety, and depression, up to three months post-discharge in patients hospitalized during the first pandemic peak with symptoms suggestive of COVID-19, to determine their HRQoL and work resumption, and to explore predictors for psychological distress and HRQoL. Additionally, we tested the hypothesis that examined whether psychological distress was more prevalent or more severe in COVID-19 confirmed patients, and in those treated in ICUs.” (Introduction, page 5, lines 88-92)

Materials and methods

 1. “Study design and setting”; when I read this heading, I expect to get information on study design and setting only. More information given here is about ethical consideration. For “study setting”, one should expect to read more of the description of the location, department, environment, infrastructure, etc… of the study place. The authors can either name this heading “study design and ethical consideration” or modify the content to suite the heading, and shift the content to an ethical heading. NB: Please read more on study setting. 

RESPONSE: 

We thank the reviewer for this useful comment. Accordingly, we have added additional information on the setting of the study:

“This period coincided with the first COVID-19 peak. During this period, several protective measures were taken in the hospital, such as the prohibition of visiting hospitalized patients and strict isolation of suspected patients until COVID-19 was ruled out.” (Methods, page 6, lines 104-106).

Additionally, we added an ‘Ethics approval”:

“Ethics approval

The study protocol was approved by the Institutional Review board of the Franciscus Gasthuis & Vlietland and deemed not to fall under the Medical Research Involving Human Subjects Act (WMO) (S1. Registered study protocol). The need for written informed consent was waived. Patients in our hospital are actively informed about the use of their anonymous data in research activities and can object against the use their data. No data were used of patients who objected against this use. The study was registered at TrialRegister.nl (registration numbeResponse: NL8882). The reporting of this study follows the Strengthening the Reporting of Observational Studies in Epidemiology (STROBE) guideline (S2. STROBE Checklist) (31).“ (Methods, page 6, lines 94-101)

2.“participants”; in this section, the inclusion and exclusion criteria are not clear. “the presence of respiratory symptoms (e.g., dyspnoea, coughing, sore throat, rhinorrhoea, saturation 24/minute) and/or gastro-intestinal symptoms (e.g., diarrhoea or vomiting) with a duration ≥24 hours”. The study inclusion and exclusion should be clear. The authors should not give examples of symptoms but should clearly state which symptoms constituted inclusion criteria and which constitute exclusion criteria. I suggest a heading “Inclusion and exclusion criteria” or Eligibility criteria. 

RESPONSE: 

We agree with the reviewer, and more clearly stated the inclusion- and exclusion criteria in the manuscript. It now reads as follows:

“Eligible patients were aged ≥18 years and hospitalized between March 16 and April 28, 2020, with symptoms suggestive of COVID-19, defined as the presence of respiratory symptoms (dyspnea, coughing, sore throat, rhinorrhoea, saturation <94% or respiratory rate >24/minute) and/or gastro-intestinal symptoms (diarrhea or vomiting) with a duration ≥24 hours, who survived until one month after hospital discharge. Patients who were unable to understand the Dutch language or did neither have a formal home address nor e-mail address were excluded.” (Methods, page 6, lines 111-116)

3. “Procedures”; With this heading two important considerations come up: (1) Data collection tools/instruments; after reading this study, 4 questionnaires and one data collection tool are supposed to be used yet none is described in this section. Please discuss briefly these tools;” (Impact 93 of Event Scale – Revised (IES-R), the Hospital Anxiety and Depression Scale (HADS), EQ-5D, RAND 36 and demographic data collection tool”

RESPONSE: 

In accordance with the academic editor’s and the previous reviewers’ comments, we have added a more detailed description of the questionnaires and how we obtained baseline demographics and treatment-related characteristics.

Authors say “Patients who were unable to understand the Dutch language or did neither have a formal home address nor e-mail address were excluded”. The dutch language is another issue on data collection tool used to collect data. It should come clearly what was the language of the questionnaires. “did the authors used a standard questionnaire in Dutch”? If yes, clarify how this was translated. If the original questionnaire was in English, clarify how it was translated in Dutch then back in English for data entry.

RESPONSE: 

We agree with the reviewer that this should be stated within the manuscript, and therefore added:

“We used validated Dutch translations of the IES-R, HADS, RAND-36, and EQ-5D.” (Methods, page 8, lines 166-167)

(2) Sampling procedure; it is not clear how the sampling procedure was done. Random sampling? Convenient sampling?.....The author needs to explain how the sampling procedure was done and clarify the sample size as well. 

RESPONSE: 

As stated in the “Participants” and “Procedures” sections of the Methods section, we invited all patients admitted with symptoms suggestive of COVID-19 to participate in the study and included patients who responded to these questionnaires. As such, convenient sampling was used. We added the following sentence into the method section:

“Convenient sampling was used. All patients admitted between March 16 and April 28, 2020, and who responded to one of the two follow-up assessments were included” (Methods, page 7, lines 125-126)

(3) It is also not clear who collected the data. Were authors collected data themselves? or trained research assistants to collect data.

RESPONSE: 

All data was collected by two researched from the study team (Johan H. Vlake and Sanne Wesselius). We added the following to the methods:

“All data was collected by members of the study team (JV and SW).” (Methods, page 8, line 167)

4.”Outcomes”; In this section, it is expected to explain how authors have reached the mentioned outcomes. I.e for EQ-5D questionnaire, the author must clarify which published algorithm used to allow a CROSSWALK to measure utility. “ The questionnaire should score using a published algorithm based on community survey in the country or neighboring country using standard Gamble, Time trade off or visual analogue scale(VAS)”. 

RESPONSE: 

We agree with the reviewer that the explanation given in the methods section regarding the outcomes measures was limited, and therefore we elaborated the description of how the sum scores, VAS score, TTO score, and summary scores were computed. In addition, we referred to the appropriate literature supporting these methods.

5.”Statistical analysis”; The analysis of this study is well done but poorly reported. i.e firstly, the author reported “ correlations between psychological outcomes and baseline/treatment characteristics were analyzed using a Spearman’s rho”. The authors should report the correlation coefficients. Secondly, the authors said “Possible predictors for psychological outcomes were analyzed using a multivariate linear regression model. Correlations with a p-value below 0.10 were included in a multivariate linear regression model to identify the predictor”. Please review this statement if this is what you wanted to say and rectify.

RESPONSE: 

We agree with the reviewer that the statistical analysis was not properly described. We revised our analysis plan and rewrote this section accordingly. We used mixed models instead of general regression models and reported al betas and p-values for the univariate and multivariate mixed models in the supplementary information.

Concordantly, we added the following:

“Differences in outcome measures between stratifications were analyzed using linear or logistic regression models for continuous and categorical outcomes, respectively, adjusting for baseline differences between groups, in which the ‘stratification’ and at baseline differing characteristics served as independent variables.” (Methods, page 9, lines 172-174).

“To identify possible predictors for psychological symptomatology or quality of life, we first conducted a univariate linear or logistic mixed model with possible predictors as independent variables (S1. Table). Variables with a p-value ≤0.10 were added in the multivariate linear or logistic mixed model to determine predictors.” (Methods, page 9, lines 180-182)

Results

1.In the participants section, the results are not clearly explained. It is hard to understand this paragraph; “A total 123 of 282 individuals participated (45%; Fig 1); 252 at one month (40%) and 201 of 365 (55%) at three months post discharge. Non-participating patients either declined participation (n=262) or did not respond (n=79)”. It is also difficult to understand the response rate of the 4 questionnaires distributed twice (at 1 month and 3 months). Please restructure this section by shading more light to this section.

RESPONSE: 

We restructured this section to make this more comprehensible:

“From March 16 to April 28, 2020, 623 patients were admitted with symptoms suggestive of COVID-19, of whom 282 patients participated (45%); 252 at one month and 201 at three months post-discharge (Fig 1). The last patient was discharged June 14, 2021, and follow-up lasted until three months after (September 14, 2020). Non-participating patient either declined participation (n=262) or did not respond to the questionnaires or to the reminding phone calls (n=79).“ (Results, page 10, lines 194-198)

In addition, we changed figure 1 to make it more comprehensible.

2. It is also not clear when the data collection was done. Please clarify these i.e; in the Study design section; the authors say “This single-centre, observational cohort study was conducted in the Franciscus Gasthuis & Vlietland hospital in Rotterdam, the Netherlands, from March 16 to September 14, 2020”. And in Participants section; the authors say “Eligible patients were aged ≥18 years and hospitalized between March 16 and April 28, 2020”. And in results section, the authors say as well “From March 16 to April 28, 2020, 623 eligible patients were admitted with symptoms suggestive of COVID-19”. 

RESPONSE: 

We understand the confusion with how we described this. Patients were included if they were admitted to the hospital between March 16, 2020, and April 28, 2020. Since the follow-up ran to three months after hospital discharge, the entire study period lasted from March 16, 2020, till three months after the hospital discharge of the last included patients, which was on June 14, 2020, and thus follow-up of the last participant ran until September 14, 2020.

To make this more understandable, we added the following sentences:

“The last patient was discharged June 14, 2020, and follow-up lasted until three months after (September 14, 2020).” (Methods, page 7, lines 124-125)

“The last patient was discharged June 14, 2021, and follow-up lasted until three months after (September 14, 2020).” (Results, page 10, lines 195-196)

 

3. My Suggestion on how the results should be presented in results section

a. Introduce the results section and present demographic characteristics in a table, comment your results from the table

RESPONSE: 

We have restructured the results, and start with the inclusion and responses of participants, followed by a section regarding demographics, in which we primarily comment on the results of Table 1, which comprise the baseline demographics and treatment-related characteristics of our cohort.

b. Present psychological distress results in a table or a clear figure, then comment the table or figure in an understandable manner. Show and comment on any P-value

RESPONSE: 

We discarded the boxplots, and present our psychological outcomes in tables only. In the results section, we focused on the most relevant data, discarded the less relevant and refer to tables for more extensive data insight.

c. Present the health related quality of life in a table or a figure, comment the results. Show and comments P-values if any 

RESPONSE: 

We have discarded the boxplots of the HRQoL and replaced them by tables. In the revised text, we focused on commenting on results which we found most relevant and referred to the tables for other study results.

d. Present the predictors of psychological outcomes and quality of life in a table showing the regression results and P-values, correlation coefficients and comment the table. 

RESPONSE: 

We have added tables regarding the predictors of the severity of psychological outcomes and predictors or HRQoL in table 4 and table 5, including betas and P-values, and only described variables that significantly predicted outcomes. The predictors for the prevalence of probable PTSD, anxiety, and depression are depicted in Table S3 in the supplementary information.

Discussion and references; I cannot comment on discussion and references for now, if results are not presented in an understandable manner

RESPONSE: 

We understand that the reviewer could not comment on the discussion at this stage. We have revised the discussion based on the revisions in the introduction, methods, and result section, and invite the reviewer to comment on the discussion section.

---

## [Decision Letter · Decision Letter 1]

9 May 2021

PONE-D-20-38679R1

Psychological distress and health-related quality of life in patients after hospitalization during the COVID-19 pandemic: a single-center, observational study.

PLOS ONE

Dear Dr. Vlake,

Thank you for submitting your manuscript to PLOS ONE. After careful consideration, we feel that it has merit but does not fully meet PLOS ONE’s publication criteria as it currently stands. Therefore, we invite you to submit a revised version of the manuscript that addresses the points raised during the review process.

PONE-D-20-38679R1 review

I have read your revised manuscript with pleasure and noticed that your manuscript improved a lot. You are one important step further. There are several things that must be addressed now before I can make a final decision. You therefore have to address the helpful comments of reviewer 3 (see below) and my specific comments below. As you will see, must comments concern clarifications, corrections and suggestion to omit certain sections. The requested revision is somewhere between a minor and major revision (PLOS ONE does not have that option). My comments are:  

Please update your literature, in intro as well as in discussion, with the very recent study by Taquet et al., 2021 (https://doi.org/10.1016/S2215-0366(21)00084-5)You examined several predictors of mental health problems and quality of life. In the introduction section explain much more in detail which predictors you want to examine and why you specifically want to examine these predictors. You can use the content of Table S1 for this purpose and omit this tables from the supp. materials. Explain too for which variables you control for when comparing groups in mental health problems and quality of life (medical conditions etc.).You examined patients from one hospital. Did you control for the possibility that patients can be identified based on the content of the Tables? If yes, add if and how the anonymity was ensured.Add Cronbach’s alphas for all scales for COVID and non-COVID patients (IES-R, etc.)Please replace the subheading Outcomes with Measures. Table 1 contains many variables that were not described in the measures section. Add them in the measures section. It may help the reader if you add sub-subheadings to the measures section (Psychological distress, Quality of life, Cause of admission, etc). In the measures section all study variables must be mentioned/explained/clarified. For instance table 2 shows that you assesses several categories psychiatric history (psychologist, etc), but this was not clarified in the text. Check all variables.Are there no cut-offs available for the HRQoL scales?Line 155. Table 1 suggest that the ICU-nonICU comparisons were conducted among the COVID patients only (39+105=144). If this is correct, please clarify this better in this line (see also below).Line 161. Please explain better the dependent variables in your analyses because there are several ways to examine potential predictors. I would like to suggest that you perform a multiple regression analyses with your distress and QOL variables at T2 as dependent variables (instead of the current analyses), and your proposed potential predictors as predictors including the corresponding distress/QOL variable at baseline. In this way you have also solved the problem that some variables were assessed at T2 (containing info over the period between T1 and T2). In addition, it offers the reader more info over the extent your variables predict distress/QOL.Line 162. What do you mean with logistic mixed model?: mixed effects logistic regression analyses?. What do you mean with “univariate linear”, correlations? (where multivariate linear = multiple regression analyses?)”. Please explain better what type of analyses you used for differences between groups and differences over time, and for predictors.Line 166. It is unclear what you meant with “Psychological outcomes of randomly selected non-responders were compared with those of responders using a linear or logistic regression model, adjusting for baseline differences between groups, in which the ‘stratification’ and at baseline differing characteristics served as independent variables”. Please clarify this better and especially “randomly selected non-responders”.  You have two type of non-response: those not participating at all (N=341) and those participating at baseline but not at follow-up. Please clarify this better in the text with results of the analyses. Move the non-response section directly above the first results section.Line 169. “All data were gathered using the International Severe Acute Respiratory and emergency Infection Consortium (ISARIC, Oxford, United Kingdom) and Franciscus Corona Registry in Castor Electronic Data Capture (Castor EDC, Amsterdam, the Netherlands).” I do not see the relationship with statistical analyses and don’t understand what you want to say here (should it be moved to the measures section in which you describe which variables were extracted from both?).Line 187. The scores and percentages in Table 1 seem an selection of Table S2. It was not clear to me how this selection was motivated. I expect that you help the reader by explaining this selection (something like “…..In table 1 the main characteristic of the study samples is presented and for a full overview of the characteristic we refer to S2 table”.Line 190. Median scores were not introduced in the stat. analyses section. The relevance of the median scores in Table 2 and 3 was totally unclear to me. What is the relevance? I would like to suggest that you conduct mixed models / ANCOVA to examine differences in sum scores of the used scales (with M and SD in tables).Line 221. You have good reasons to refer to supp. materials because no significant associations were found. However you can inform the reader about this (similar at other places) in a more gentle way (for instance:  “”Analyses showed that none of the assessed predictors were significantly associated with probable PTSD, probable anxiety, and probable  depression (for the results we therefore refer to ……)”.Line 221. “natural decline” what do you mean with natural? It is a decline, and we do not know from your data that it is natural.Table 1. Please insert zero’s (N and %) instead of blanks because blanks may suggest that a variable was not assessed among a subgroup.Table 2. The total numbers at T1 and T2 in this table do not match figure 1. Check or explain differences (or add info in figure).In table 1 the ICU and nonICU numbers at baseline are 39 and 105. However in Table 2 at base line they are 32 and 216 (!). This is very confusing. Please correct and revise. (In Table 3 numbers are missing).For the analyses of the differences between the subgroups, you controlled for differences at baseline. Please clarify how many control variables were entered in each analyses (present also event-per-variable ratio) so readers can see that your analyses are justified (EVP higher than 10 as a rule of thumb).Line 246. Work resumption. Again, this topic was hardly introduced in the intro (what dod we know from the literature about work resumption after  hospitalization). Perhaps it is an idea to omit this topic from your manuscript.Table 4. This table suggest that IES-R and HADS scores at T1 were not predictive. Is this correct? (this is a very remarkable outcome).Discussion. To help the reader please use more subheadings in your discussion section.Line 268. See comment 1. I believe some rewriting is needed because of this study by Taquet et al. 2021.Line 285. Ref. 41 is about life time prevalence (and retrospective!) and should not be used in this way.  The prevalence do not seem very high compared to victims of f.i. accidents, medical errors, and (sexual) violence (see van der velden et al. 2020. https://doi.org/10.1371/journal.pone.0232477)Line 331. “Baseline psychological status and HRQoL in our cohort was not available”. You mean before infection/hospitalization? (you used the word baseline to describe T1).Line 350-353. Should “Physicians should be aware of the psychological consequences of hospitalization during a pandemic but not only in those affected by COVID-19 or requiring ICU treatment” be “However, Physicians should be aware of the psychological consequences of hospitalization during a pandemic but not only in those affected by COVID-19 or requiring ICU treatment”.Upload all supplementary materials as separate files.

We look forward to receiving your revised manuscript.

Kind regards,

Peter G. van der Velden, Ph.D.

Academic Editor

PLOS ONE

Journal Requirements:

Reviewers' comments:

Reviewer's Responses to Questions

**Comments to the Author**

1. If the authors have adequately addressed your comments raised in a previous round of review and you feel that this manuscript is now acceptable for publication, you may indicate that here to bypass the “Comments to the Author” section, enter your conflict of interest statement in the “Confidential to Editor” section, and submit your "Accept" recommendation.

Reviewer #2: (No Response)

Reviewer #3: (No Response)

2. Is the manuscript technically sound, and do the data support the conclusions?

Reviewer #2: Yes

Reviewer #3: Yes

3. Has the statistical analysis been performed appropriately and rigorously? 

Reviewer #2: Yes

Reviewer #3: No

4. Have the authors made all data underlying the findings in their manuscript fully available?

Reviewer #2: Yes

Reviewer #3: Yes

5. Is the manuscript presented in an intelligible fashion and written in standard English?

Reviewer #2: Yes

Reviewer #3: Yes

6. Review Comments to the Author

Reviewer #2: The authors have sufficiently addressed my previous concerns.

Congratulations on the interesting paper.

Reviewer #3: Second review: Psychological distress and health-related quality of life in patients after hospitalization during the COVID-19 pandemic: a single-center, observational study

I want to thank the authors for the good job done, especially on methodology. There is a significant improvement on the quality of the manuscript. However, some work need to be done to allow the paper to be published

Introduction

1. This section was reviewed partially. “This paper’s aim is to quantify psychological distress and to determine their HRQoL”. The introduction needs a section to briefly discuss few different methods to assess mental health and HRQoL. This section will help non expert readers to understand and also give a good flavor to the reader

Methodology

Authors have made significant progress in term of reporting methodology. However, data analysis part is not well reported;

it is not clear in the analysis which method used at each level, for example;

2. Line 172; “Differences in outcome measures between stratifications were analyzed using linear or logistic regression”. Clarify where you used linear and where you used logistic regression and clarify the logistic model used (i.e function or Equation). For logistic regression (binary outcomes), I was expecting to read the results with Odds ratios and confidence intervals. Please explain this!

3. Line 185-186 “Psychological outcomes of randomly selected non-responders were compared with those of responders using a linear or logistic regression model”. It is not clear where applied linear regression, where applied multiple and where applied logistic regression. Please clarify these by showing which statistical model used (i.e function or equation).

4. Please discuss with your analyst if there is a possibility of using “ Time to event outcomes regression” (cox proportional hazards model) and calculate hazard ratios for ICU versus non-ICU and COVID-19 versus non COVID-19 in the development or reduction of psychological distress from the time of discharge to one month time follow up!

Health related quality of life;

5. Clarify how you calculated utility scores and health states, then how did you do the crosswalk for the health states?

6. Explain how you calculated the mean utility scores

7. Explain how you estimated QALYS lost due to psychological distress

Results

All relevant or significant results (i. e significant P-values) in the tables in this section must be briefly commented below each table.

8. For example, “Table1: Baseline demographics and treatment-related characteristics of study population”. Where are comments of this table?

9. Table2 “Psychological outcomes throughout follow-up” is confusing because it is not commented. I don’t see the relevance of this table. And if am not wrong, no significant results are reported in this table. And then, the bottom of this table reads logistic regression. With logistic regression results, one would expect to be reported Odds ratios (ORs) and confidence intervals. I suggest you remove this table because the results are confusing and are not explained therefore it does not add value unless if it is clearly commented.

10. Table3, 4 and 5 are relevant but are not commented. Please comment important or significant results in these tables

Discussion

I am still not able to comment on discussion; results need to be worked on before reviewing the discussion part.

11. Intext citations and references don’t follow PLOS ONE guidelines. Please check on PLOS ONE referencing guidelines

7. PLOS authors have the option to publish the peer review history of their article (what does this mean?). If published, this will include your full peer review and any attached files.

Reviewer #2: **Yes: **Mark Bosmans

Reviewer #3: **Yes: **Desire Aime Nshimirimana

---

## [Author Response · Author response to Decision Letter 1]

22 Jun 2021

Response to the handling editor:

Dear Dr. Vlake,

Thank you for submitting your manuscript to PLOS ONE. After careful consideration, we feel that it has merit but does not fully meet PLOS ONE’s publication criteria as it currently stands. Therefore, we invite you to submit a revised version of the manuscript that addresses the points raised during the review process.

I have read your revised manuscript with pleasure and noticed that your manuscript improved a lot. You are one important step further. There are several things that must be addressed now before I can make a final decision. You therefore have to address the helpful comments of reviewer 3 (see below) and my specific comments below. As you will see, must comments concern clarifications, corrections and suggestion to omit certain sections. The requested revision is somewhere between a minor and major revision (PLOS ONE does not have that option).

Response:

Dear Peter van der Velden,

We really appreciate that you are currently handling our manuscript and that you have given us the opportunity to thoroughly revise our manuscript. 

The suggestions, concerns, and comments raised by you and reviewer 3 did further improve our manuscript and we have made point-by-point responses to your and the reviewer’s comments below.

For the referral to certain pages or lines within the manuscript; all referrals are based on the submitted marked-up manuscript, in which the changes made during the revision are highlighted.

My comments are: 

1. Please update your literature, in intro as well as in discussion, with the very recent study by Taquet et al., 2021 (https://doi.org/10.1016/S2215-0366(21)00084-5)

RESPONSE: 

We agree with the editor that an update of published articles is in place for both the introduction and discussion sections. We therefore updated the introduction and discussion, as suggested by the editor.

We added the following text into the introduction and discussion section:

“The uncertainties and measures surrounding the pandemic raised concerns of increased psychological distress in non-hospitalized citizens (20). Since measures taken in hospital were more drastic, concerns of the mental well-being of hospitalized patients were even higher (21, 22). Data from the previous SARS and MERS epidemics support this concern and early studies suggest that up to 50% of COVID-19 patients suffer from psychological distress up to two months after hospital discharge (23-28). More recently, Taquet et al. estimated that the risk for psychiatric sequelae is higher in COVID-19 patients and in those admitted to ICU using electronic health records data. Complementary observational studies are needed to add direction as to whether these risks may be attributed to COVID-19 or hospitalization during a pandemic, and corroborate on the different risks between specific subsets of patients, such as non-COVID-19 and ICU patients (29).” (Introduction, page 3, lines 63-71)

“In a recent large retrospective study, COVID-19 was indeed associated with a higher risk of psychiatric outcomes as compared to influenza and respiratory tract infections (29). In our cohort however, psychological distress was largely similar between non-COVID-19 patients with respiratory symptoms and COVID-19 patients, and we did not observe COVID-19 positivity to be a predictor for severity of PTSD, anxiety, or depression. As such, our data collectively suggest that COVID-19 on its own has no major influence on psychological outcome. The contradictory findings of the Taquet and our study will largely be explained by the difference in study design (retrospective cohort vs. prospective observational cohort), the different outcome measures used (ICD-based diagnoses vs. questionnaire-based outcome) and comparing the covid-19 cohort with other control populations (influenza and respiratory tract infection vs. patient with symptoms suggestive of COVID-19 but COVID-19 PCR negatives).“ (Discussion, page 25, lines 369-378)

2. You examined several predictors of mental health problems and quality of life. In the introduction section explain much more in detail which predictors you want to examine and why you specifically want to examine these predictors. You can use the content of Table S1 for this purpose and omit this tables from the supp. materials. Explain too for which variables you control for when comparing groups in mental health problems and quality of life (medical conditions etc.).

RESPONSE: 

We agree with the editor, and added the following to the introduction concerning the predictors examined:

“Illnesses requiring hospitalization, particularly those requiring intensive care unit (ICU) treatment, are known to negatively impact post-discharge psychological well-being; symptoms of depression and anxiety occur in up to 67% of hospitalized patients and symptoms of post-traumatic stress disorder (PTSD) in up to 45% (2-15). This post-hospitalization psychological distress negatively impacts the health-related quality of life (HRQoL) (7, 11, 12, 14). Both demographic characteristics, such as female gender, lower educational level, unemployment and non-western ethnicity, and treatment-related characteristics, such as duration of admission and severity of the disease, are known to be negatively associated with psychological well-being of patients (2, 8, 16-19). 

 The uncertainties and measures surrounding the pandemic raised concerns of increased psychological distress in non-hospitalized citizens (20). Since measures taken in hospital were more drastic, concerns of the mental well-being of hospitalized patients were even higher (21, 22). Data from the previous SARS and MERS epidemics support this concern and early studies suggest that up to 50% of COVID-19 patients suffer from psychological distress up to two months after hospital discharge (23-28). More recently, Taquet et al. estimated that the risk for psychiatric sequelae is higher in COVID-19 patients and in those admitted to ICU using electronic health records data. Complementary observational studies are needed to add direction as to whether these risks may be attributed to COVID-19 or hospitalization during a pandemic, and corroborate on the different risks between specific subsets of patients, such as non-COVID-19 and ICU patients (29).” (Introduction, page 3, lines 56-71)

We however believe that Table S1 has added value for the reader who is interested in an overview of the literature supporting our choice of predictors, and we therefore would like to suggest to keep Table S1 as part of the Supplements.

With regard to the controlling variables in the comparison between COVID-19 and non-COVID-19 patients, and the comparison between COVID-19 ICU and COVID-19 non-ICU patients; firstly, we better described this in the Statistical analysis section of the methods:

“Differences in outcome measures between stratifications were analyzed using simple linear or logistic regression models, to adjust for at baseline differing characteristics, for continuous and categorical outcomes, respectively. We therefore first performed a simple univariate linear (for continuous outcomes) or logistic (for categorical outcomes) regression analyses, in which all at baseline differing characteristics were analyzed one-by-one. Variables that were associated with the outcome, i.e., a p-value ≤0.10, were added as independent variables to the simple multivariate regression models and were as such adjusted for.“ (Methods, Statistical analysis, page 9, lines 195-200)

Additionally, we described for which variables we adjusted in the legends of the Tables.

3. You examined patients from one hospital. Did you control for the possibility that patients can be identified based on the content of the Tables? If yes, add if and how the anonymity was ensured.

RESPONSE: 

We understand the concerns of the editor. We have re-checked whether the content of our tables could lead to the identification of individual patients in our study and omitted categorical variables from the supplementary dataset that were only present in one patients of a stratifications and could as such lead to the identification of an individual patient. These included ethnicity, psychiatric history outcomes, treatment code before discharge, and presence of several comorbidities. 

For other tables; we only present the median values or regression outcomes of the entire population and stratifications. In none of the stratifications, nor in the entire cohort, probable PTSD, anxiety, or depression was reported by only 1 patient. As such, patients cannot be identified based on these tables.

4. Add Cronbach’s alphas for all scales for COVID and non-COVID patients (IES-R, etc.)

RESPONSE: 

We agree, and added the following for the Statistical analysis section of the methods:

“The internal reliability of all questionnaires used (IES-R, HADS, EQ-5D and RAND-36) were analyzed using Cronbach’s alpha and all showed a high internal reliability (Table S2).” (Methods, Statistical analysis, page 9, line 190-191)

Additionally, we added a table depicting the Cronbach’s alphas to the supplements:

5. Please replace the subheading Outcomes with Measures. Table 1 contains many variables that were not described in the measures section. Add them in the measures section. It may help the reader if you add sub-subheadings to the measures section (Psychological distress, Quality of life, Cause of admission, etc). In the measures section all study variables must be mentioned/explained/clarified. For instance table 2 shows that you assesses several categories psychiatric history (psychologist, etc), but this was not clarified in the text. Check all variables.

RESPONSE: 

We apologize for the untidiness. We have now described all outcomes that were collected during the study, replaced the subheading Outcomes with Measures, and added subheadings to the section.

The section now reads:

“Measures

All data were gathered using the International Severe Acute Respiratory and emergency Infection Consortium (ISARIC, Oxford, United Kingdom) and Franciscus Corona Registry in Castor Electronic Data Capture (Castor EDC, Amsterdam, the Netherlands). The ISARIC database in an international initiative to collect baseline demographics and treatment-related characteristics of all patients admitted to the hospital with respiratory symptoms during the SARS-CoV-2 pandemic. The Franciscus Corona Registry is a local addition to the ISARIC database, in which variables that were not collected in the ISARIC database, but were required for the current trial, were collected. All data was collected by members of the study team (JV and SW).

Baseline demographics and treatment-related characteristics

The following baseline demographics were retrieved from electronic healthcare records: age (years), ethnicity (Caucasian, black, Surinamese/Hindustan, Arab (not specified), Turkish, Moroccan, others, or unknown), sex at birth (male or female), body mass index (kg / m2), comorbidities (yes/no: hypertension, chronic cardiac disease, chronic pulmonary disease, asthma, tuberculosis, chronic kidney disease, mild liver disease, moderate liver disease, chronic neurological disease, dementia, chronic hematologic disease, diabetes type I or II, rheumatologic disorder, malignant neoplasm; total number of comorbidities), smoking status (yes, never smoked, former smoker, unknown). Patients were asked about their educational level (i.e., elementary school, high school, intermediate vocational education, higher professional education, university education) and employment characteristics (employed before hospitalization yes/no, weekly work hours before admission, weekly work hours after discharge, healthcare worker yes/no). Additionally, we asked patients about their mental history, i.e., whether they had encountered psychological impairments in the past 5 years and whether they were treated for these impairments by a psychologist, psychiatrist or had received medication. Patients were free to decide whether or not to answer the questions regarding their mental history.

 The following treatment-related characteristics were retrieved from electronic healthcare records: cause of admission, treatment restrictions at the day of hospital admission, last registered treatment restriction before discharge, hospital length of stay, ICU admission (yes/no, length of stay), SOFA score at admission, P/F ratio at admission, S/F ratio at admission, oxygen therapy (yes/no, duration), non-invasive ventilation (yes/no, duration), invasive ventilation (yes/no, duration), prone positioning (yes/no, duration), tracheostomy (yes/no) and survival during follow-up.

Primary outcome: Psychological distress

The primary outcome was the prevalence and severity of probable PTSD, depression, and anxiety, assessed using validated Dutch translations of the Impact of Event Scale – Revised (IES-R) and the Hospital Anxiety and Depression Scale (HADS) at one and three months post-discharge. 

 The IES-R is a 22-item questionnaire that quantifies the subjective distress a person is experience after a traumatic event (31). The IES-R yields a sum score, ranging from 0-88 (higher scores indicating more severe symptoms), and subscale scores can be calculated for symptoms of intrusion, avoidance, and hyperarousal. The IES-R sum score was considered as the severity of PTSD, and an IES-R sum score above 24 was defined as probable PTSD (32). 

 The HADS is commonly used to determine the levels of anxiety and depression that a person is experiencing (33). The HADS is a 14-item scale that generates ordinal data. Seven of the items relate to anxiety and seven relate to depression. The HADS yields a depression and anxiety sum score, ranging from 0 to 21, with higher scores indicating more severe symptoms. The HADS anxiety and depression score will be considered as the severity of anxiety- and depression-related symptoms, respectively. A HADS depression or anxiety score above 8 was defined as probable depression or probable anxiety, respectively (33, 34). 

Secondary outcome: Health-related quality of life

The secondary outcome was HRQoL. HRQoL was assessed using validated Dutch translations of the EuroQoL 5-dimensions-5-levels (EQ-5D) and the RAND-36 questionnaires at one and three months post-discharge. 

 The EQ-5D measures the HRQoL on five dimensions (mobility, self-care, usual activities, pain/discomfort, and anxiety/depression), by which the weight of a health state can be computed into an EQ-5D Time Trade Off (TTO) score. This score ranges from – 0.446 (worst quality of life) to 1.000 (best quality of life) and will be considered as the overall HRQoL. Additionally, patients score their current subjective perceived health state on a visual analogue scale (EQ-5D VAS), ranging from 0 (worst health imaginable) to 100 (best health imaginable) (35, 36). Based on the distribution of age in our cohort, the mean TTO score of the Dutch general population is 0.852 (36). A TTO score below 0.852 is considered poor, and a TTO score above 0.852 good. 

 The RAND-36 is a 36-item, self-reported survey of HRQoL, and consists nine scales scores, which are the weighted sums of the questions in their section (37). Each scale is directly transformed to a scale ranging from 0 (worse score) to 100 (best score) on the assumption that each question carries an equal weight. The nine scores are: physical functioning, social functioning, physical role limitations, emotional role limitations, mental health, vitality, pain, general health, and health change. Based on these scores, a mental and physical component score (MSC-36, PCS-36) can be computed for which the mean in the general population will be 50 with a standard deviation (SD) of 10. The MCS-36 score will be considered as the mental HRQoL and the PCS-36 will be considered as the physical HRQoL. A MCS-36 or PCS-36 score below 50 is considered low, and a MCS or PCS score above 50 is considered good (38). 

Exploratory outcomes

We additionally explored predictors for the severity and prevalence of psychological distress and the HRQoL. These predictors were chosen based on previous literature, and included: age, gender, ethnicity, educational level, duration of admission, ICU admission, COVID-19 diagnosis and work before admission, and severity of disease in terms of SOFA score at the first day of COVID-19 suspicion (day of enrolment) (2, 8, 16-19). Literature supporting these predictors is depicted in Table S1 in the Supporting Information.” (Methods, Measures, page 6-8, lines 109-176)

6. Are there no cut-offs available for the HRQoL scales?

RESPONSE: 

The editor is right that there are no cut-offs available for the HRQoL scales. As the EQ-5D and RAND-36 are used to quantify HRQoL, rather than measure the presence of a disease, there are no pre-specified cut-off values available. Therefore, we used the mean values of the MCS-36, PCS-36, and RAND-36 subscales of the general Dutch population, retrieved from validation studies, as reference.

7. Line 155. Table 1 suggest that the ICU-nonICU comparisons were conducted among the COVID patients only (39+105=144). If this is correct, please clarify this better in this line (see also below).

RESPONSE: 

The editor is right that we compared outcomes of COVID ICU and COVID-19 ICU survivors. To explain: only 2 patients were admitted to the ICU with another cause than COVID-19. As such, we decided to analyze differences between ICU and non-ICU patients only within the COVID-19 cohort.

To clarify this matter, we described this in the first rows of Table 1.

8. Line 161. Please explain better the dependent variables in your analyses because there are several ways to examine potential predictors. I would like to suggest that you perform a multiple regression analyses with your distress and QOL variables at T2 as dependent variables (instead of the current analyses), and your proposed potential predictors as predictors including the corresponding distress/QOL variable at baseline. In this way you have also solved the problem that some variables were assessed at T2 (containing info over the period between T1 and T2). In addition, it offers the reader more info over the extent your variables predict distress/QOL.

RESPONSE: 

We thank the editor for this comment, and in hindsight understand that we did not explain thoroughly enough how we conducted our analysis. 

First, we want to stress that the T1 measurement was not a baseline measurement, but the first follow-up assessment at one month post-hospital discharge. As such, using the psychological distress and QoL outcomes at T1 as independent variables in our regression models will firstly disable us to draw any conclusions about the course of psychological distress, i.e., whether psychological distress improved or worsened during follow-up, and secondly will possibly disable us to find relevant risk factors, as these are expected to be correlated to the outcomes at T1 and therefore are likely to lose significance in the model.

Also, with regard to the concerns about the missing data; in our analysis, we used mixed effects linear regression models. These account for both inter-group differences, such as the stratifications and at baseline differing variables between stratifications, and intra-group differences, such as time after discharge. Using mixed effects linear regression models, slopes are computed for each individual patient, and the coefficient computed takes into account all of these slopes, and all data provided. As such, the model uses all data available instead of only participants who filled out both follow-up assessments, and thereby adjusts for missing values. Hence, we believe that the current analysis is more robust than the suggested analysis of the editor.

To clarify, we added the following to the Statistical analysis section of the Methods:

“To identify possible predictors for the severity and prevalence of psychological distress and the overall, mental, and physical HRQoL, we first conducted univariate mixed effects regression analysis: a linear model for continuous outcomes and a logistic model for categorical outcomes. We used mixed effects regression models to adjust for intergroup (i.e., time) and intragroup differences (i.e., cohort and variables of interest). In these, the possible predictive variable served as independent variable one-by-one and the outcome of interest at both follow-up time points served as dependent variables. Secondly, all variables which showed a p-value ≤0.10 in the univariate mixed effects regression model were added to the multivariate mixed effects regression model to determine which variables significantly predicted the outcome. We report the coefficient [95% CI], which implies the estimated mean difference, for linear models, and odds ratios (ORs), including its 95% CI, for logistic models.” (Methods, Statistical Analysis, page 9-10, lines 201-209)

9. Line 162. What do you mean with logistic mixed model?: mixed effects logistic regression analyses?. What do you mean with “univariate linear”, correlations? (where multivariate linear = multiple regression analyses?)”. Please explain better what type of analyses you used for differences between groups and differences over time, and for predictors.

RESPONSE: 

We apologize for the unclarity, and partially answered these questions in your previous comment. In the univariate mixed effect linear/logistic regression model, we assessed whether any association was observed between the possible predictor and the outcome of interest. Thereafter, all variables that showed a high correlation, i.e., a p-value ≤0.10, were added to the multivariate mixed effects regression model. Variables that showed a significant contribution to the multivariate mixed effects regression model were considered as independent predictors.

10. Line 166. It is unclear what you meant with “Psychological outcomes of randomly selected non-responders were compared with those of responders using a linear or logistic regression model, adjusting for baseline differences between groups, in which the ‘stratification’ and at baseline differing characteristics served as independent variables”. Please clarify this better and especially “randomly selected non-responders”. You have two type of non-response: those not participating at all (N=341) and those participating at baseline but not at follow-up. Please clarify this better in the text with results of the analyses. Move the non-response section directly above the first results section.

RESPONSE: 

We thank the editor for this comment, as we realize we did not explain clearly enough how we assessed the non-responders bias.

To clarify, to analyze non-responders bias, we asked non-responders, i.e., patients who had expressed a wish not to take part of this study or who had not returned any of the questionnaire, to fill out a single set of questionnaires approximately 1 month after the last questionnaire was sent. A total of 12 non-responders agreed, and the outcomes of these 12 non-responders were compared with the outcomes of ‘true’ responders to determine whether there were (substantial) differences in outcomes.

We agree that we did not formulate with clear enough, and changed this accordingly:

“We randomly approached non-responders (who did not respond at both time-points) four months after hospital discharge, i.e., one month after sending the last questionnaire, and asked them to fill out a single set of questionnaires to analyze non-responder’s bias.” (Methods, Procedures, page 6, lines 106-108)

“Baseline characteristics of all non-responders were compared with those of responders using a Mann-Whitney U test for continuous variables and using a Fisher’s exact test for categorical variables. Psychological outcomes of randomly selected non-responders (who did not respond at both time-points) were compared with those of responders, and psychological outcomes of full responders were compared with those of partial responders (i.e., patients who only responded at one or three months), using simple linear or logistic regression models for continuous and categorical variables, respectively. In these, we adjusted for variables that were expected to confound the outcome and differed at baseline between stratifications as described above.” (Statistical analysis, methods, page 10, lines 210-216)

Additionally, we moved the ‘Non-responder’s analysis’ section to the beginning of the result section.

11. Line 169. “All data were gathered using the International Severe Acute Respiratory and emergency Infection Consortium (ISARIC, Oxford, United Kingdom) and Franciscus Corona Registry in Castor Electronic Data Capture (Castor EDC, Amsterdam, the Netherlands).” I do not see the relationship with statistical analyses and don’t understand what you want to say here (should it be moved to the measures section in which you describe which variables were extracted from both?).

RESPONSE: 

We agree and changed accordingly, and moved this to the measures section:

“All data were gathered using the International Severe Acute Respiratory and emergency Infection Consortium (ISARIC, Oxford, United Kingdom) and Franciscus Corona Registry in Castor Electronic Data Capture (Castor EDC, Amsterdam, the Netherlands). The ISARIC database in an international initiative to collect baseline demographics and treatment-related characteristics of all patients admitted to the hospital with respiratory symptoms during the SARS-CoV-2 pandemic. The Franciscus Corona Registry is a local addition to the ISARIC database, in which variables that were not collected in the ISARIC database, but were required for the current trial, were collected.” (Methods, Measures, page 6, lines 110-115)

12. Line 187. The scores and percentages in Table 1 seem an selection of Table S2. It was not clear to me how this selection was motivated. I expect that you help the reader by explaining this selection (something like “…..In table 1 the main characteristic of the study samples is presented and for a full overview of the characteristic we refer to S2 table”.

RESPONSE: 

The editor is right that Table 1 is a selection of Table S2, and changed accordingly:

“Table 1 depicts the most relevant baseline demographics and treatment-related characteristics; a full overview of all collected demographics and characteristics can be found in Table S8.“ (Results, Participants, page 11, line 238-239)

13. Line 190. Median scores were not introduced in the stat. analyses section. The relevance of the median scores in Table 2 and 3 was totally unclear to me. What is the relevance? I would like to suggest that you conduct mixed models / ANCOVA to examine differences in sum scores of the used scales (with M and SD in tables).

RESPONSE: 

We apologize for this untidiness, and now described in the Statistical analysis section how variables will be presented:

“Continuous variables are presented as mean, including its standard deviation (SD), if normally distributed, and as median, including its 95% range, if not normally distributed. Categorical variables are presented as absolute and relative frequency.” (Methods, Statistical analysis, page 9, lines 178-180)

With regard to Table 2 and 3; we regret that the relevance of these tables was unclear, and a similar comment was raised by reviewer 3. Table 2 and 3 depicts our primary and secondary outcome measures; Table 2 represents the severity and prevalence of psychological distress, as measured at both follow-up moments. The P-values provided regarding the comparison between stratifications are calculated using simple linear and logistic regression models, which is largely comparable with an ANCOVA, to adjust for at baseline differing variables. The P-values regarding the change over time were calculated using mixed effects linear and logistic regression models. To clarify, we added the coefficients of the regression models to the Tables.

As the outcome variables presented in these tables were not normally distributed, we report medians rather than means. We can however still use regression models as long as the residuals of the model are normally distributed, and no other assumptions are violated.

To clarify this, we revised the legend of Table 2:

“Descriptive statistics of the psychological distress outcomes, stratified by COVID-19 diagnosis and ICU admission. Severity of PTSD, anxiety, and depression were expressed as the IES-R, HADS anxiety, and HADS depression sum scores, respectively. Prevalence of probable PTSD, anxiety, and depression was defined as the proportion of patients scoring above the cut-off. Abbreviations: CI, confidence interval; COVID-19, coronavirus disease 2019; ICU, intensive care unit; OR, odds ratio; PTSD, post-traumatic stress disorder. Differences over time were calculated using mixed effects linear (for continuous outcomes) and logistic (for categorical outcomes) regression models, with time as independent variable. Differences between stratifications were analyzed using simple linear (for continuous outcomes) and logistic (for categorical outcomes) regression models.“ (Results, Table 2, page 16, line 272)

And changed the legend of Table 3 accordingly:

“Descriptive statistics of the HRQoL outcomes, stratified by COVID-19 diagnosis and ICU admission. Differences over time were calculated using mixed effects linear (for continuous outcomes) and logistic (for categorical outcomes) regression models, with time as independent variable. Abbreviations: CI, confidence interval; COVID-19, coronavirus disease 2019; EQ-5D, European quality of life 5 dimensions questionnaire; HRQoL, health-related quality of life; ICU, intensive care unit; MCS-36, mental component score of the RAND-36; OR, odds ratio; PCS-36, physical component score of the RAND-36; TTO, time trade-off; VAS, visual analogue scale. Differences between stratifications were analyzed using simple linear (for continuous outcomes) and logistic (for categorical outcomes) regression models.” (Results, Table 3, page 18, line 299)

14. Line 221. You have good reasons to refer to supp. materials because no significant associations were found. However you can inform the reader about this (similar at other places) in a more gentle way (for instance: “”Analyses showed that none of the assessed predictors were significantly associated with probable PTSD, probable anxiety, and probable depression (for the results we therefore refer to ……)”.

RESPONSE: We agree and changed accordingly:

“Table 4 and S9 depict the results of the exploration of predictors for the severity and prevalence of probable PTSD, anxiety and depression. ” (Results, Predictors of psychological outcomes and quality of life, page 17, line 292-293)

“None of the predictors were significantly associated with the prevalence of probable PTSD (Table S9).” (Results, Predictors of psychological outcomes and quality of life, page 17, lines 297-298)

“None of the predictors were significantly associated with the prevalence of probable anxiety (Table S9).” (Results, Predictors of psychological outcomes and quality of life, page 19, line 304-305)

“None of the predictors were significantly associated with the prevalence of probable depression (Table S9).“ (Results, Predictors of psychological outcomes and quality of life, page 19, lines 311-312)

15. Line 221. “natural decline” what do you mean with natural? It is a decline, and we do not know from your data that it is natural.

RESPONSE: 

We agree and omitted natural throughout the manuscript.

16. Table 1. Please insert zero’s (N and %) instead of blanks because blanks may suggest that a variable was not assessed among a subgroup.

RESPONSE: 

We changed Table 1 accordingly.

17. Table 2. The total numbers at T1 and T2 in this table do not match figure 1. Check or explain differences (or add info in figure).

RESPONSE: 

We apologize for the untidiness and checked all numbers. The numbers now all match with those in Figure 1.

18. In table 1 the ICU and nonICU numbers at baseline are 39 and 105. However in Table 2 at base line they are 32 and 216 (!). This is very confusing. Please correct and revise. (In Table 3 numbers are missing).

RESPONSE: 

We apologize for the untidiness and corrected the number.

19. For the analyses of the differences between the subgroups, you controlled for differences at baseline. Please clarify how many control variables were entered in each analyses (present also event-per-variable ratio) so readers can see that your analyses are justified (EVP higher than 10 as a rule of thumb).

RESPONSE: 

We understand the concerns of the reviewer. In hindsight, we realize that we have indeed adjusted for too many characteristics. We therefore now first analyzed the association between at baseline differing characteristics using simple univariate linear (for continuous outcomes) or logistic (for categorical outcomes) regression models. In these, the outcome variable served as dependent variable, and the at baseline differing characteristics one-by-one as independent variables. All at baseline differing characteristics which showed a high association, i.e., a p-value ≤0.10, were used in simple multivariate linear (for continuous outcomes) or logistic (for categorical outcomes) regression models, to compare outcomes between stratifications, while adjusting for at baseline differing characteristics which could confound the outcome. 

Using this method, the number of variables for which was controlled did not violate the event-per-variable ratio.

We described this in the statistical analysis section of the methods:

“Differences in outcome measures between stratifications were analyzed using simple linear or logistic regression models, to adjust for at baseline differing characteristics, for continuous and categorical outcomes, respectively. We therefore first performed a simple univariate linear (for continuous outcomes) or logistic (for categorical outcomes) regression analyses, in which all at baseline differing characteristics were analyzed one-by-one. Variables that were associated with the outcome, i.e., a p-value ≤0.10, were added as independent variables to the simple multivariate regression models and were as such adjusted for.” (Methods, statistical analysis, page 9, lines 195-200)

20. Line 246. Work resumption. Again, this topic was hardly introduced in the intro (what dod we know from the literature about work resumption after hospitalization). Perhaps it is an idea to omit this topic from your manuscript.

RESPONSE: 

We agree with the editor and omitted this topic from the manuscript.

21. Table 4. This table suggest that IES-R and HADS scores at T1 were not predictive. Is this correct? (this is a very remarkable outcome).

RESPONSE:

The outcomes of the IES-R and HADS, both at 1 month (T1) and 3 months (T2) were both added as dependent variables, we therefore added time as independent variable. Table 4 depicts the outcomes of the mixed-effects linear regression model, in which the IES-R sum score, HADS anxiety score, and HADS depression score at both follow-up time-points served as dependent variables (using a longfile), while time and at baseline differing variables (see our answer to your previous comment) served as independent variables. As such, no conclusions can be drawn about a possible predictive nature of psychological outcomes at T1 from Table 4, as we did not examine this.

22. Discussion. To help the reader please use more subheadings in your discussion section.

RESPONSE: 

We agree with the editor and added the subheadings “Limitations” and “Conclusions” to our discussion section. 

Although we fully agree with the editor that subheadings in a discussion may be of help to guide the reader through the discussion, in this discussion we feel that adding additional subheadings feels rather artificial and rather distracts than helps the reader.

23. Line 268. See comment 1. I believe some rewriting is needed because of this study by Taquet et al. 2021.

RESPONSE: 

We agree and added the following:

“In a recent large retrospective study, COVID-19 was indeed associated with a higher risk of psychiatric outcomes as compared to influenza and respiratory tract infections (29). In our cohort however, psychological distress was largely similar between non-COVID-19 patients with respiratory symptoms and COVID-19 patients, and we did not observe COVID-19 positivity to be a predictor for severity of PTSD, anxiety, or depression. As such, our data collectively suggest that COVID-19 on its own has no major influence on psychological outcome. The contradictory findings of the Taquet and our study will largely be explained by the difference in study design (retrospective cohort vs. prospective observational cohort), the different outcome measures used (ICD-based diagnoses vs. questionnaire-based outcome) and comparing the covid-19 cohort with other control populations (influenza and respiratory tract infection vs. patient with symptoms suggestive of COVID-19 but COVID-19 PCR negatives).” (Discussion, page 25, lines 369-378)

24. Line 285. Ref. 41 is about life time prevalence (and retrospective!) and should not be used in this way. The prevalence do not seem very high compared to victims of f.i. accidents, medical errors, and (sexual) violence (see van der velden et al. 2020. https://doi.org/10.1371/journal.pone.0232477)

RESPONSE: 

We apologize for the untidiness, and changed accordingly:

“The observed prevalence of psychological distress in our cohort appears to be higher than in the general population during non-pandemic circumstances, and was similar to people suffering from PTSD-related symptoms after a traumatic event, such as theft, burglary, accidents and death of a significant other (41).” (Discussion, page 24, lines 354-356)

25. Line 331. “Baseline psychological status and HRQoL in our cohort was not available”. You mean before infection/hospitalization? (you used the word baseline to describe T1).

RESPONSE: 

We agree and changed accordingly. This sentence now reads:

“In our cohort, psychological status and HRQoL prior to hospitalization were not available and thus we cannot formally rule out that pre-existing psychological well-being was poorer, predisposing for impaired psychological recovery (47).” (Discussion, page 25, lines 380-382)

“First, the high incidence of psychological symptomatology and the poor HRQoL post-discharge may be either attributed to factors related to hospitalization or to baseline psychological imbalances, as psychological status and HRQoL of participants prior to hospitalization was not available.” (Discussion, page 26, lines 407-410)

We however want to stress that T1 is not described as baseline, as T1 refers to the first follow-up time-point, which was one month after discharge from the hospital.

26. Line 350-353. Should “Physicians should be aware of the psychological consequences of hospitalization during a pandemic but not only in those affected by COVID-19 or requiring ICU treatment” be “However, Physicians should be aware of the psychological consequences of hospitalization during a pandemic but not only in those affected by COVID-19 or requiring ICU treatment”.

RESPONSE: 

We agree and changed accordingly.

27. Upload all supplementary materials as separate files.

RESPONSE: 

We have uploaded all supplementary materials as separate files as requested by the editor.

Response to Reviewer #3: 

Second review: Psychological distress and health-related quality of life in patients after hospitalization during the COVID-19 pandemic: a single-center, observational study

I want to thank the authors for the good job done, especially on methodology. There is a significant improvement on the quality of the manuscript. However, some work need to be done to allow the paper to be published

Introduction

1. This section was reviewed partially. “This paper’s aim is to quantify psychological distress and to determine their HRQoL ”. The introduction needs a section to briefly discuss few different methods to assess mental health and HRQoL. This section will help non expert readers to understand and also give a good flavor to the reader

RESPONSE: 

First, we thank the reviewer for his useful comments and suggestions.

With regard to the suggestion to briefly discuss few different methods to assess mental health and HRQoL, we believe that elaborating more on this matter will limit the focus and conciseness of the introduction, and does, in our opinion, not add value to the introduction.

We however understand that the choices made regarding the assessment of psychological distress and HRQoL remains a matter of debate, and therefore discussed this, although briefly in the discussion section:

“Fourth, we assessed psychological well-being and HRQoL using self-report questionnaires. Although commonly used and extensively validated, formal assessment of psychologic disorders requires consultation with a psychologist or psychiatrist, and usage of self-reports may result in an overestimation of psychologic distress.” (Discussion, page 26, lines 420-422)

Methodology

Authors have made significant progress in term of reporting methodology. However, data analysis part is not well reported;

it is not clear in the analysis which method used at each level, for example;

2. Line 172; “Differences in outcome measures between stratifications were analyzed using linear or logistic regression”. Clarify where you used linear and where you used logistic regression and clarify the logistic model used (i.e function or Equation). For logistic regression (binary outcomes), I was expecting to read the results with Odds ratios and confidence intervals. Please explain this!

RESPONSE: 

First, we apologize for the unclarity. A similar comment was raised by the academic editor. To clarify; we used simple linear regression models to compare continuous outcomes between stratifications, and simple logistic regression models to compare dichotomous/categorical variables. Mixed effects linear regression models were used to analyze the course of psychological symptomatology, in means of the IES-R sum score, HADS anxiety score, and HADS depression score, and the course of HRQoL, in means of mental quality of life (MCS-36), physical quality of life (PCS-36), and overall HRQoL (EQ-5D TTO score), and we used mixed effect logistic regression models to analyze the course of the prevalence of probable PTSD, anxiety, and depression, as defined as patients scoring above the described cut-off value. Furthermore, for the identification of risk factors for psychological distress and HRQoL, we used mixed effects linear (for continuous outcomes) or logistic (for categorical outcomes) regression models.

We changed the description of this in the Statistical analysis section of the Methods:

“Differences in outcome measures between stratifications were analyzed using simple linear or logistic regression models, to adjust for at baseline differing characteristics, for continuous and categorical outcomes, respectively. We therefore first performed a simple univariate linear (for continuous outcomes) or logistic (for categorical outcomes) regression analyses, in which all at baseline differing characteristics were analyzed one-by-one. Variables that were associated with the outcome, i.e., a p-value ≤0.10, were added as independent variables to the simple multivariate regression models and were as such adjusted for.

 To identify possible predictors for the severity and prevalence of psychological distress and the overall, mental, and physical HRQoL, we first conducted univariate mixed effects regression analysis: a linear model for continuous outcomes and a logistic model for categorical outcomes. We used mixed effects regression models to adjust for intergroup (i.e., time) and intragroup differences (i.e., cohort and variables of interest). In these, the possible predictive variable served as independent variable one-by-one and the outcome of interest at both follow-up time points served as dependent variables. Secondly, all variables which showed a p-value ≤0.10 in the univariate mixed effects regression model were added to the multivariate mixed effects regression model to determine which variables significantly predicted the outcome.” (Methods, Statistical analysis, page 9-10, lines 195-208)

Also, the reviewer rightfully points out that it was inappropriate to report the raw coefficients of the logistic regression models. We have changes these to OR’s and the corresponding confidence intervals:

“We report the coefficient [95% CI], which implies the estimated mean difference, for linear models, and odds ratios (ORs), including its 95% CI, for logistic models.” (Methods, Statistical analysis, page 10, lines 208-209)

3. Line 185-186 “Psychological outcomes of randomly selected non-responders were compared with those of responders using a linear or logistic regression model”. It is not clear where applied linear regression, where applied multiple and where applied logistic regression. Please clarify these by showing which statistical model used (i.e function or equation).

RESPONSE: 

We apologize for the untidiness, and are convinced that we now better described how non-responders were analyzed:

“Baseline characteristics of all non-responders were compared with those of responders using a Mann-Whitney U test for continuous variables and using a Fisher’s exact test for categorical variables. Psychological outcomes of randomly selected non-responders (who did not respond at both time-points) were compared with those of responders, and psychological outcomes of full responders were compared with those of partial responders (i.e., patients who only responded at one or three months), using simple linear or logistic regression models for continuous and categorical variables, respectively. In these, we adjusted for variables that were expected to confound the outcome and differed at baseline between stratifications as described above.” (Methods, Statistical analysis, page 10, lines 210-216)

4. Please discuss with your analyst if there is a possibility of using “ Time to event outcomes regression” (cox proportional hazards model) and calculate hazard ratios for ICU versus non-ICU and COVID-19 versus non COVID-19 in the development or reduction of psychological distress from the time of discharge to one month time follow up!

RESPONSE: 

We appreciate the suggestion of the reviewer and discussed this matter with our statistician (B. Boxma-de Klerk). Unfortunately, such an analysis requires us to know at what time point exactly someone started suffering from psychological distress, rather than if a patient reported psychological distress at a certain follow-up assessment. In our study, we only assessed psychological distress at two time-points: one month and three months after discharge. For patients who already reported psychological distress at one point, or for those who did not reported psychological distress at one month, but did at three months, we cannot know when during follow-up these symptoms first manifested. As such, it is not possible to use time to event outcomes regression.

Health related quality of life;

5. Clarify how you calculated utility scores and health states, then how did you do the crosswalk for the health states?

R:ESPONSE 

The time trade off (TTO) score of the EQ-5D is computed by giving a weight for eight possible answers for each item. These weights are subtracted from the maximum; which is 1. So, every patients starts with a EQ-5D TTO score of 1.00. For each answer given on one of the five domains, a certain weight is subtracted. For instance, for the mobility domains, no subtraction is conducted when a patients has no problems walking, 0.035 is subtracted when a patient has slight problems walking, 0.057 is subtracted when a patient has moderate problems with walking, 0.166 is subtracted when a patients has severe problems with walking, and 0.203 is subtracted when a patients is unable to walk. Every answer possibility has other weights, and an additional weight is subtracted when a patient reports the worse score on all five domains. These weights are country-specific, and published within the validation study of the EQ-5D in the Dutch population. 

The computation of the EQ-5D utility/TTO score is described in the paper of Herdman et al. (2011) and Versteegh et al. (2016), to which we refer when we first mention the EQ-5D . 

With regard to the MCS-36 and PCS-36; the calculations for these summary scales are extensively described in the paper of Ware et al. (1994). In summary; first all subscales are calculated, and subsequently these are normalized against the general population, so that the mean is 50 and the standard deviation is 10 in the general population. The mean of the subscales in its section represent the mental and physical component score.

As the EQ-5D and RAND-36 are the most commonly used instrument to determine a person’s quality of life, and the exact calculations of the health states/summary scales does not fall under the scope of the present study, we do not further elaborate on this within the manuscript.

6. Explain how you calculated the mean utility scores

RESPONSE: 

We have explained the calculations behind the utility scores in the response to your previous comment.

7. Explain how you estimated QALYS lost due to psychological distress

RESPONSE: 

We do not fully comprehend the reviewer’s comment, as we do not estimate and discuss any loss of QALYs due to psychological distress in the manuscript under review. 

Results

All relevant or significant results (i. e significant P-values) in the tables in this section must be briefly commented below each table.

RESPONSE: 

We indeed did not comment on all significant p-values within the manuscript to maintain conciseness. 

Based on the comment of the reviewer, we added the following to the manuscript to address all significant outcomes of Table 1:

“Table 1 depicts the most relevant baseline demographics and treatment-related characteristics; a full overview of all collected demographics and characteristics can be found in Table S8. The median age was 64 years (95% range 33-88), 106 (36%) were female, and a history of mental illness was reported by 27 of 201 patients (13%), who were willing to share this information. The median hospital length of stay (LOS) was 4 days (1-52) and 42 patients (14%) required ICU admission (median ICU LOS; 16 days [95% range 0-52]). Overall, 146 patients (50%) were SARS-CoV-2 positive. Non-COVID-19 patients were predominantly diagnosed with other pulmonary or cardiac illnesses (Table S8). COVID-19 patients were younger, had fewer comorbidities, more frequently worked prior to hospitalization, had a longer hospital and ICU LOS, were more frequently admitted to the ICU, had a higher SOFA score and lower P/F ratio on admission, more frequently received oxygen therapy and invasive ventilation for a longer duration, were more frequently mechanically ventilated in prone position and more frequently received a tracheotomy than non-COVID-19 patients (Table 1). Within COVID-19 patients, those who were admitted to the ICU had a longer hospital LOS, a higher SOFA score and lower P/F ratio, received oxygen therapy more frequently and more frequently received a tracheostomy (Table 1).” (Results, Participants, page 12-14, lines 238-251).

We had already commented on all significant results from Table 2 and 3.

Based on the comment of the reviewer, we added the following to the manuscript to address all significant outcomes of Table 4 and 5:

“Time after discharge (p=0.01), not being Caucasian (p<0.01) and having completed higher professional education were associated with the severity of PTSD in the univariate mixed effects regression analyses and were included in the multivariate mixed effects regression analysis. Of these,…” (Results, Predictors of psychological distress and quality of life, page 17, lines 293-295)

“Time after discharge (p=0.01), female gender (p=0.01), having completed higher professional education (p=0.01), employment status before hospitalization (p=0.08), and COVID-19 diagnosis (p=0.06) were associated with the severity of anxiety in the univariate mixed effects regression analyses and were included in the multivariate mixed effects regression analysis. Of these,…“ (Results, Predictors of psychological distress and quality of life, page 19, lines 300-303)

“Time after discharge (p=0.01), female gender (p=0.02), not being Caucasian (p=0.06), having completed higher professional (p<0.01) or university education (p=0.01), employment status before hospitalization (p=0.02), COVID-19 diagnosis (p=0.03) and ICU admission (p=0.07) were associated with the severity of depression in the univariate mixed effects regression analyses and were included in the multivariate mixed effects regression analysis. Of these,…” (Results, Predictors of psychological distress and quality of life, page 19, lines 306-310)

“Time after discharge (p=0.01), age (p<0.01), female gender (p<0.01), having completed higher professional (p<0.01) or university education (p<0.01) and COVID-19 diagnosis (p<0.01) were associated with overall HRQoL in the univariate mixed effects regression analyses and were included in the multivariate mixed effects regression analysis. Of these,…” (Results, Predictors of psychological distress and quality of life, page 19, lines 313-316)

“Time after discharge (p<0..01), female gender (p<0.01), being not Caucasian (p=0.05), having completed university education (p=0.03), being mechanically ventilated (p=0.03), ICU admission (p=0.01) and having more severe symptoms of PTSD (p<0.01), anxiety (p<0.01) and depression (p<0.01) were associated with mental HRQoL in the univariate mixed effects regression analyses and were included in the multivariate mixed effects regression analysis. Of these,…” (Results, Predictors of psychological distress and quality of life, page 19, lines 321-325)

“Lastly, time after discharge (p<0.01), age (p<0.01), female gender (p<0.01), employment status before hospitalization (p<0.01), hospital LOS (p=0.07), COVID-19 diagnosis (p<0.01) and having more severe symptoms of PTSD (p<0.01), anxiety (p<0.01) and depression (p<0.01) were associated with physical HRQoL in the univariate mixed effects regression analyses and were included in the multivariate mixed effects regression analysis. Of these,… “ (Results, Predictors of psychological distress and quality of life, page 20, lines 328-331)

8. For example, “Table1: Baseline demographics and treatment-related characteristics of study population”. Where are comments of this table?

RESPONSE: 

We have described all variables that were differing at baseline in the results section, as described in the answer to your previous comment.

9. Table2 “Psychological outcomes throughout follow-up” is confusing because it is not commented. I don’t see the relevance of this table. And if am not wrong, no significant results are reported in this table. And then, the bottom of this table reads logistic regression. With logistic regression results, one would expect to be reported Odds ratios (ORs) and confidence intervals. I suggest you remove this table because the results are confusing and are not explained therefore it does not add value unless if it is clearly commented.

RESPONSE: 

We regret that the relevance of these tables was unclear, and a similar comment was raised by the academic editor. Table 2 and 3 depicts our primary and secondary outcome measures; Table 2 represents the severity and prevalence of psychological distress, as measured at both follow-up moments. The P-values provided regarding the comparison between stratifications are calculated using simple linear and logistic regression models, which is largely comparable with an ANCOVA, to adjust for at baseline differing variables, and the P-values regarding the change over time were calculated using mixed effects linear and logistic regression models. 

To clarify this, we revised the legend of Table 2:

“Descriptive statistics of the psychological distress outcomes, stratified by COVID-19 diagnosis and ICU admission. Severity of PTSD, anxiety, and depression were expressed as the IES-R, HADS anxiety, and HADS depression sum scores, respectively. Prevalence of probable PTSD, anxiety, and depression was defined as the proportion of patients scoring above the cut-off. Abbreviations: CI, confidence interval; COVID-19, coronavirus disease 2019; ICU, intensive care unit; OR, odds ratio; PTSD, post-traumatic stress disorder. Differences over time were calculated using mixed effects linear (for continuous outcomes) and logistic (for categorical outcomes) regression models, with time as independent variable. Differences between stratifications were analyzed using simple linear (for continuous outcomes) and logistic (for categorical outcomes) regression models.“ (Results, Table 2, page 16, line 272)

And changed the legend of Table 3 accordingly:

“Descriptive statistics of the HRQoL outcomes, stratified by COVID-19 diagnosis and ICU admission. Differences over time were calculated using mixed effects linear (for continuous outcomes) and logistic (for categorical outcomes) regression models, with time as independent variable. Abbreviations: CI, confidence interval; COVID-19, coronavirus disease 2019; EQ-5D, European quality of life 5 dimensions questionnaire; HRQoL, health-related quality of life; ICU, intensive care unit; MCS-36, mental component score of the RAND-36; OR, odds ratio; PCS-36, physical component score of the RAND-36; TTO, time trade-off; VAS, visual analogue scale. Differences between stratifications were analyzed using simple linear (for continuous outcomes) and logistic (for categorical outcomes) regression models.” (Results, Table 3, page 18, line 299)

10. Table3, 4 and 5 are relevant but are not commented. Please comment important or significant results in these tables

RESPONSE: 

We understand the confusion of the reviewer, as we indeed did not comment on the univariate analysis depicted in these tables. As described in your previous comment, we have now commented on all significant results in these tables.

Discussion

I am still not able to comment on discussion; results need to be worked on before reviewing the discussion part.

11. Intext citations and references don’t follow PLOS ONE guidelines. Please check on PLOS ONE referencing guidelines

RESPONSE: 

We apologize for the untidiness, and fully copy-edited the references section so that all references follow the PLOS ONE guidelines.

---

## [Decision Letter · Decision Letter 2]

26 Jul 2021

Psychological distress and health-related quality of life in patients after hospitalization during the COVID-19 pandemic: a single-center, observational study.

PONE-D-20-38679R2

Dear Dr. Vlake,

We’re pleased to inform you that your manuscript has been judged scientifically suitable for publication and will be formally accepted for publication once it meets all outstanding technical requirements.

Kind regards, en van harte gefeliciteerd,

Peter G. van der Velden, Ph.D.

Academic Editor

PLOS ONE

Additional Editor Comments (optional):

Reviewers' comments:

Reviewer's Responses to Questions

**Comments to the Author**

1. If the authors have adequately addressed your comments raised in a previous round of review and you feel that this manuscript is now acceptable for publication, you may indicate that here to bypass the “Comments to the Author” section, enter your conflict of interest statement in the “Confidential to Editor” section, and submit your "Accept" recommendation.

Reviewer #2: All comments have been addressed

Reviewer #3: All comments have been addressed

2. Is the manuscript technically sound, and do the data support the conclusions?

Reviewer #2: Yes

Reviewer #3: Yes

3. Has the statistical analysis been performed appropriately and rigorously? 

Reviewer #2: Yes

Reviewer #3: Yes

4. Have the authors made all data underlying the findings in their manuscript fully available?

Reviewer #2: (No Response)

Reviewer #3: Yes

5. Is the manuscript presented in an intelligible fashion and written in standard English?

Reviewer #2: Yes

Reviewer #3: Yes

6. Review Comments to the Author

Reviewer #2: (No Response)

Reviewer #3: This paper has now reached to the publication level. I thank the authors for the job well done. Congratulations!

7. PLOS authors have the option to publish the peer review history of their article (what does this mean?). If published, this will include your full peer review and any attached files.

Reviewer #2: **Yes: **Mark Bosmans

Reviewer #3: **Yes: **Desire Aime Nshimirimana

---

## [Editor Report · Acceptance letter]

30 Jul 2021

PONE-D-20-38679R2 

Psychological distress and health-related quality of life in patients after hospitalization during the COVID-19 pandemic: a single-center, observational study. 

Dear Dr. Vlake:

I'm pleased to inform you that your manuscript has been deemed suitable for publication in PLOS ONE. Congratulations! Your manuscript is now with our production department. 

Kind regards, 

on behalf of

Dr. Peter G. van der Velden 

Academic Editor

PLOS ONE